# Cell cycle constraints on capsulation and bacteriophage susceptibility

Silvia Ardissone[1], Coralie Fumeaux[1], Matthieu Bergé[1], Audrey Beaussart[2], Laurence Théraulaz[1], Sunish Kumar Radhakrishnan[1†], Yves F Dufrêne[2], Patrick H Viollier[1]*

[1]Department of Microbiology and Molecular Medicine, Institute of Genetics and Genomics in Geneva, University of Geneva, Geneva, Switzerland; [2]Institute of Life Sciences, Université catholique de Louvain, Louvain-la-Neuve, Belgium

**Abstract** Despite the crucial role of bacterial capsules in pathogenesis, it is still unknown if systemic cues such as the cell cycle can control capsule biogenesis. In this study, we show that the capsule of the synchronizable model bacterium *Caulobacter crescentus* is cell cycle regulated and we unearth a bacterial transglutaminase homolog, HvyA, as restriction factor that prevents capsulation in G1-phase cells. This capsule protects cells from infection by a generalized transducing *Caulobacter* phage (φCr30), and the loss of HvyA confers insensitivity towards φCr30. Control of capsulation during the cell cycle could serve as a simple means to prevent steric hindrance of flagellar motility or to ensure that phage-mediated genetic exchange happens before the onset of DNA replication. Moreover, the multi-layered regulatory circuitry directing HvyA expression to G1-phase is conserved during evolution, and HvyA orthologues from related *Sinorhizobia* can prevent capsulation in *Caulobacter*, indicating that alpha-proteobacteria have retained HvyA activity.

*For correspondence: patrick.viollier@unige.ch

**Present address:** †School of Biology, Indian Institute of Science Education and Research, Thiruvananthapuram, Thiruvananthapuram, India

**Competing interests:** The authors declare that no competing interests exist.

**Reviewing editor**: Gisela Storz, National Institute of Child Health and Human Development, United States

## Introduction

Genetic exchange is both fundamental to the adaptation of bacterial cells faced with ever-changing environmental conditions and the cause of the alarming dissemination of antibiotic resistance determinants among the bacterial pathogens. The underlying mechanisms include direct uptake of naked DNA (transformation) by bacterial cells as well as cell- or bacteriophage-based delivery systems (respectively conjugation and generalized transduction) (*Wiedenbeck and Cohan, 2011*; *Seitz and Blokesch, 2013*). Thus, uncovering mechanisms that curb genetic exchange could provide new entry points to help intervene with the spread of antibiotic resistances. While genetic exchange can be facilitated in response to changes in the number of cells in a population (quorum sensing) or other developmental states (*Seitz and Blokesch, 2013*), an important but yet unresolved question is whether genetic exchange can also be regulated by systemic cues, such as those directing cell cycle progression. Recent cytological experiments provide evidence that components of the pneumococcal natural transformation (competence) machinery can be linked to cell division, at least spatially (*Bergé et al., 2013*), hinting that unknown mechanisms may indeed restrict genetic exchange in time or in space during the progression of the cell division cycle. A myriad of events are coordinated with progression through the eukaryotic cell cycle, but our understanding of such mechanisms and the factors that constrain them during the bacterial cell cycle are sparse.

Microbial polysaccharidic capsules can also restrict bacteriophage-mediated genetic exchange. Typically, they mask bacteriophage receptor sites that are on or near the cell surface (*Hyman and Abedon, 2010*). Moreover, capsules are virulence factors in many Gram-negative and Gram-positive species, as they provide immune evasion by shielding or camouflaging the targets of host immune

**eLife digest** Many bacteria have a tough outer coating known as capsule that protects them from untoward environmental conditions. This capsule also prevents viruses called bacteriophages from invading the bacterial cells, and it shields those bacteria that can infect humans from attack by our immune system. External conditions—such as a lack of nutrients and physical stresses—are known to trigger capsule formation. However, almost nothing is known about the signals from within the bacteria that control the formation of a capsule.

Now, Ardissone et al. have used the capsulated bacterium called *Caulobacter crescentus* to show that capsule formation is regulated by the bacterial cell cycle. This cycle is a series of events and checkpoints that happen every time a cell divides to form two new cells. Ardissone et al. revealed that capsule cannot form during the first phase of the cell cycle. The bacterium only forms its capsule as this phase ends and before it copies its DNA and later divides in two.

Ardissone et al. discovered that an enzyme called HvyA, which is only produced during the first phase of the cell cycle, prevents the capsule from forming. Inactivating the HvyA enzyme was also shown to make the bacteria impervious to infection by a bacteriophage. Furthermore, Ardissone et al. dissected the complicated steps involved in regulating the production of the HvyA enzyme and showed that such regulatory steps are also used by other species of bacteria.

Without their capsules, bacteria can take up new genetic material from a number of sources that might help them adapt to a changing environment. Ardissone et al.'s findings suggest that by only exchanging genetic material during the first phase of the cell cycle, bacteria ensure that any useful DNA is taken up and copied along with their own DNA later in the cell cycle.

Antibiotic resistance spreads between bacteria via the exchange of genetic material, making it increasingly difficult to treat bacterial infections. Interfering with the formation of the capsule during an infection could help overcome this problem by making the bacteria more vulnerable to attack either by our own immune system or by bacteriophages that can be used to treat bacterial infections. By investigating how genetic exchange and capsule formation are linked and regulated, the findings of Ardissone et al. might now open up new strategies to help combat bacterial infections.

cells that are located on the surface of bacterial cells (*Schneider et al., 2007*; *Kadioglu et al., 2008*).

While capsulation can be regulated by nutritional cues (*Kadioglu et al., 2008*; *Yother, 2011*), cell envelope stresses (*Laubacher and Ades, 2008*) or physical cues (*Sledjeski and Gottesman, 1996*; *Tschowri et al., 2009*; *Loh et al., 2013*), no systemic cues are currently known. As virulence regulators have recently been found to control bacterial cell cycle transcription (*Fumeaux et al., 2014*), capsulation might also be regulated by the cell cycle. The synchronizable and capsulated alpha-proteobacterium *Caulobacter crescentus* (henceforth *Caulobacter*) is the pre-eminent model system for cell cycle studies (*Ravenscroft et al., 1991*; *Skerker and Laub, 2004*; *Curtis and Brun, 2010*). It recently transpired that many of the emerging concepts of cell cycle control, and the underlying mechanisms such as those directing an asymmetric cell division (*Hallez et al., 2004*), are also operational in other alpha-proteobacterial lineages (*Kobayashi et al., 2009*; *Brilli et al., 2010*; *Ardissone and Viollier, 2012*; *Pini et al., 2013*; *Fumeaux et al., 2014*). In *Caulobacter* this asymmetric cell division yields a motile and piliated swarmer (SW) cell that is in a G1-arrested state and a sessile stalked (ST) cell that resides in S-phase (*Figure 1A*). The cellular buoyancy of the latter is higher than the former, a feature that has been exploited for synchronization of *Caulobacter* populations (by density gradient centrifugation) (*Ely, 1991*) for cell cycle studies. A vast number of transcripts are cell cycle-regulated in *Caulobacter* (*Laub et al., 2000*) and in the alpha-proteobacterium *Sinorhizobium meliloti* (*De Nisco et al., 2014*), a plant symbiont. Importantly, many transcripts of orthologous genes are cell cycle-regulated and this is in large part governed by the conserved and essential cell cycle transcriptional regulator A (CtrA). CtrA activates transcription of many late S- and G1-phase genes that are repressed by the transcriptional regulators SciP or the MucR1/2 paralogs, respectively (*Fumeaux et al., 2014*; *Gora et al., 2010*; *Tan et al., 2010*). In addition to acting as a transcriptional activator, CtrA functions as a negative regulator of gene expression and DNA replication initiation by binding to the conserved 5'-TTAA-(N)$_7$-TTAA-3' motif (CtrA box) located in many *Caulobacter* and *Sinorhizobium*

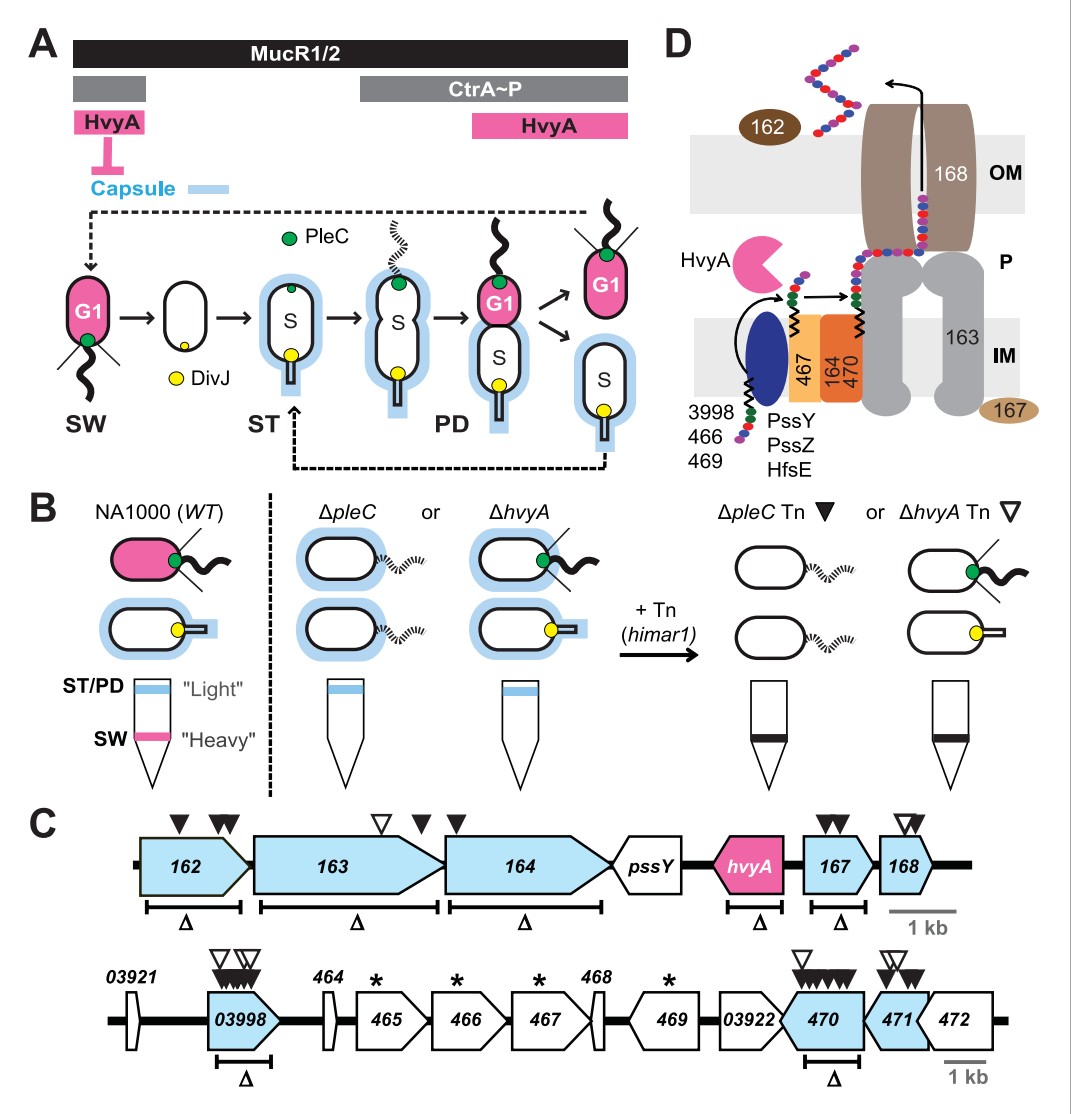

**Figure 1**. Capsulation of *Caulobacter* cells is cell cycle regulated. (**A**) Schematic of the *Caulobacter* cell cycle and the regulatory interactions that determine the presence/absence of the capsule (in blue). Phosphorylated CtrA (CtrA~P) and MucR1/2 control expression of *hvyA*. The antagonistic kinase/phosphatase pair, DivJ (yellow dot) and PleC (green dot), indirectly influences CtrA~P and partitions with the stalked (ST) cell chamber or swarmer (SW) cell chamber, respectively. PleC promotes CtrA~P accumulation in the SW cell. HvyA prevents encapsulation in SW cells. Pink denotes HvyA accumulation in the SW (G1) cell compartment. Light blue indicates the presence of the capsule in ST(S) and pre-divisional (PD) cells. (**B**) Schematic of cell buoyancy upon centrifugation on density gradient for *WT Caulobacter* cells (left). SW cells sediment in the lower band ('heavy', in pink) whereas ST and PD cells sediment in the upper band ('light', in blue). Δ*pleC* and Δ*hvyA* cells are 'light' due to the constitutive presence of capsule (middle). Upon transposon mutagenesis with *himar1* Tn, we isolated 'heavy' non-capsulated mutants by cell density centrifugation (right). (**C**) *Caulobacter* loci identified by the cell density screen. The upper panel represents the *CCNA_00162-CCNA_00168* locus and the lower panel the *CCNA_03921-CCNA_00472* locus on the mobile genetic element of *Caulobacter* NA1000. The fragment deleted for each in-frame deletion is indicated (Δ). Black triangles indicate Tn insertions obtained in the Δ*pleC* background and white triangles indicate Tn insertions obtained in the Δ*hvyA* background. *CCNA_00166* (*hvyA*) is shown in pink, and the genes hit by our buoyancy screen for 'heavy' mutants are in blue. The asterisks show the ORFs identified as essential by ***Christen et al. (2011)***. (**D**) Schematic of capsule polymerisation/export system based on the one for group 1 CPS in *E. coli* (***Collins and Derrick, 2007***). Putative functions were attributed to *Caulobacter* proteins based on the homology and conserved domains. CCNA_03998, CCNA_00466, and CCNA_00469 are putative glycosyltransferases; PssY, PssZ, and HfsE are polyisoprenylphosphate hexose-1-phosphotransferases; CCNA_00467 is a putative

*Figure 1. Continued on next page*

*Figure 1. Continued*

flippase; CCNA_00165 and CCNA_00470 are polymerases; CCNA_00163 and CCNA_00167 have homology to the tyrosine autokinase Wzc and its phosphatase Wzb, respectively, that regulate polymerisation and export; CCNA_00168 is the putative outer membrane lipoprotein required for translocation of the polysaccharide across the outer membrane.

promoters (*Laub et al., 2000*, *2002*; *Fumeaux et al., 2014*) and the *Caulobacter* origin of replication (*Cori*) (*Quon et al., 1998*).

Binding of CtrA to its target sites is stimulated 100-fold by phosphorylation of aspartate at position 51 (D51, CtrA~P) (*Siam and Marczynski, 2000*) through a complex phosphorelay that controls both abundance and phosphorylation of CtrA as a function of the cell cycle (*Biondi et al., 2006*; *Iniesta et al., 2006*). CtrA~P is present in G1-phase, proteolyzed during the G1→S transition to permit replication initiation and re-accumulates later in S-phase (*Domian et al., 1997*).

G1-phase transcription is also positively dependent on PleC (*Wang et al., 1993*), a phosphatase that is sequestered to the SW cell pole at cell division (*Wheeler and Shapiro, 1999*). While PleC also regulates the cellular buoyancy properties, the molecular basis has never been determined. In this study, we use suppressor genetics to unearth the *Caulobacter* capsule as determinant of the buoyancy trait and we identify HvyA, a member of the poorly characterized bacterial transglutaminase-like cysteine protease (BTLCP) family, as a PleC-dependent negative regulator that is restricted to G1-phase to prevent capsulation at this time in the cell cycle. As the capsule protects *Caulobacter* cells from infection by the generalized transducing Caulophage φCr30 and no CRISPR/Cas (clustered regularly interspaced short palindromic repeats–CRISPR associated)-based adaptive immunity system to protect cells from invading genetic material is encoded in the *Caulobacter* genome (*Marks et al., 2010*), HvyA is the first example of a factor restricting phage infection to a confined cell cycle phase.

## Results

### A putative capsule export machinery governs the buoyancy switch

The switch in cellular buoyancy in NA1000 (*WT*) *Caulobacter* cells is cell cycle-regulated, but the underlying regulatory mechanism is elusive. We used a developmental mutant (Δ*pleC*) as an entry point to identify the genetic determinants conferring the change in buoyancy. Density gradient centrifugation of a *WT* culture yields ST and PD cells with the characteristic high buoyancy (for simplicity, we refer to these cells as 'light') and SW cells with the characteristic low buoyancy ('heavy' cells, *Figure 1B*). As Δ*pleC* cells are exclusively 'light', we simply sought *himar1* transposon (Tn) insertions that render Δ*pleC* cells 'heavy' (*Figure 1B*). After backcrossing such 'heavy' Δ*pleC*::Tn mutants, we mapped the Tn insertion sites to two loci (*Figure 1C*).

The first locus (*Figure 1C*, upper panel) encodes putative components of a group 1 (Wzy)-like capsular polysaccharide (CPS) export machinery, in which the saccharide precursors are first assembled on undecaprenol (Und~P, black zigzag in *Figure 1D*) on the cytoplasmic membrane, flipped and assembled in the periplasm into a polymer that is then translocated across the outer membrane and anchored on the cell surface (*Whitfield, 2006*). Tn insertions were found in the genes encoding a putative capsular polysaccharide biosynthesis lipoprotein (*CCNA_00162*), a Wzc-like chain length regulator/tyrosine kinase (*CCNA_00163*), a putative O-antigen polymerase/ligase (*CCNA_00164*), a putative Wzb-like metallophosphatase (*CCNA_00167*), and a Wza-like outer membrane translocon (*CCNA_00168*), all commonly associated with capsular export systems. No Tn insertions were found in the other two genes within this cluster, *CCNA_00166* and *pssY*. For the latter, this could be explained by a functional redundancy of *pssY* with the orthologs encoded by *pssZ* and *hfsE*, all encoding poly-isoprenylphosphate hexose-1-phosphotransferases (*Toh et al., 2008*). By contrast, *CCNA_00166* is predicted to encode a putative bacterial transglutaminase-like cysteine protease (BTLCP) and we describe below that an in-frame deletion of *CCNA_00166* (Δ*hvyA* in *Figure 1C*) in *WT* cells resulted in 'light' cells, akin to Δ*pleC* cells. Consistent with the results from the Tn analysis, an in-frame deletion (*Figure 1C*) in *pssY* did not render cells 'heavy'. By contrast, deleting *CCNA_00162*, *CCNA_00163*, *CCNA_00164*, or Δ*CCNA_00167* gave rise to 'heavy' *WT* (and Δ*pleC*) cells (*Figure 1C* and *Figure 2A*, *Supplementary file 1*). Moreover, complementation of Δ*CCNA_00162*, Δ*CCNA_00163*, Δ*CCNA_00164*, Δ*CCNA_00167*, or *CCNA_00168*::Tn mutant cells with a plasmid harbouring the corresponding gene

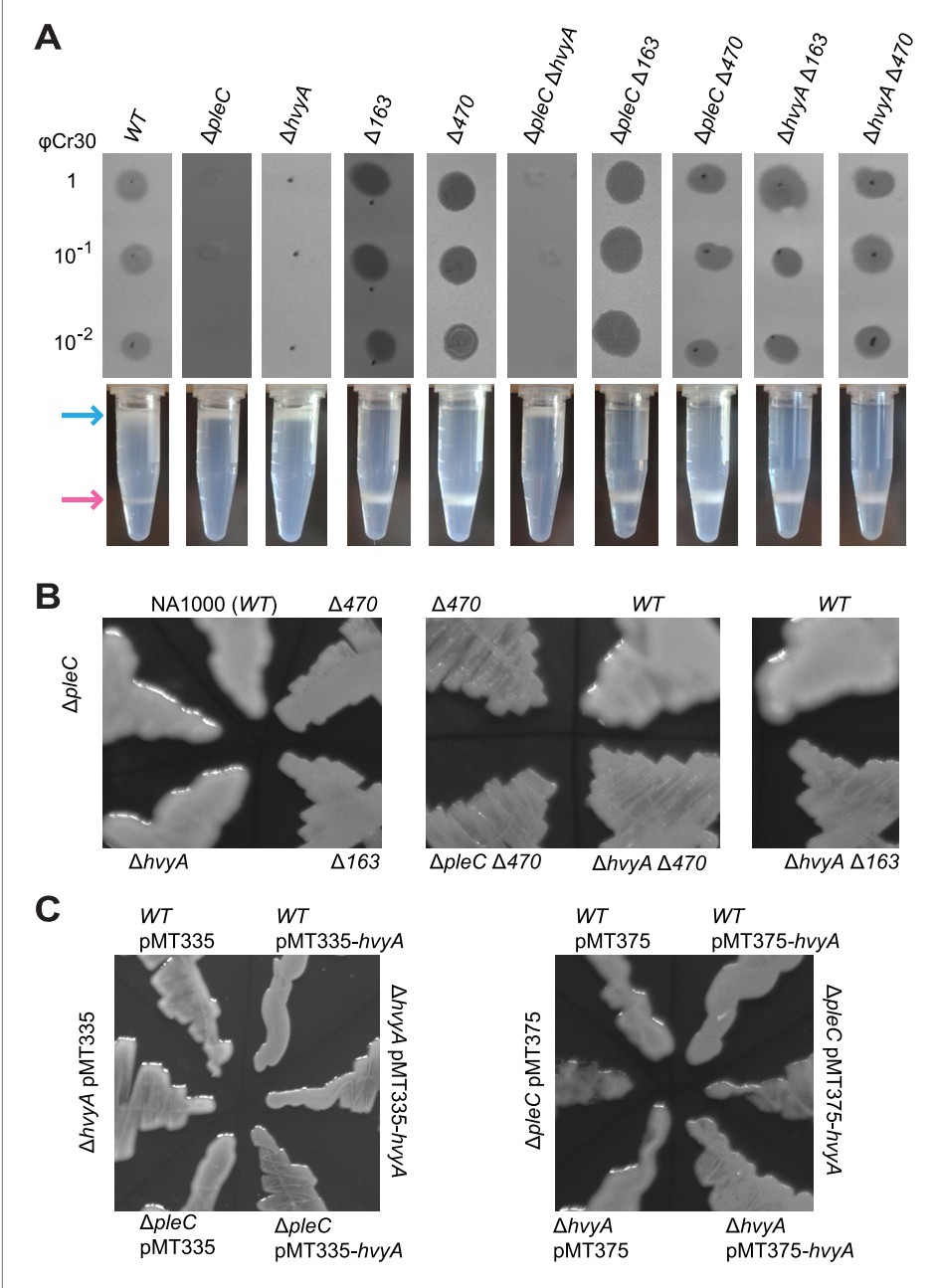

**Figure 2**. Capsulation affects buoyancy, mucoidy, and bacteriophage sensitivity. (**A**) Sensitivity to bacteriophage φCr30 and buoyancy of *Caulobacter WT* (NA1000) and different mutant strains. Mutation in *CCNA_00163* or *CCNA_00470* restores sensitivity to φCr30 in Δ*pleC* and Δ*hvyA* mutant backgrounds. Δ*pleC*, Δ*hvyA*, and the double mutant Δ*pleC* Δ*hvyA* are 'light', whereas mutation in *CCNA_00163* or *CCNA_00470* renders cells 'heavy' (also in a Δ*pleC* or Δ*hvyA* background). (**B**) Mucoidy of *Caulobacter WT* (NA1000) and different mutant strains plated on PYE medium supplemented with 3% sucrose. *WT*, Δ*pleC*, and Δ*hvyA* are highly mucoid, whereas mutations in *CCNA_00163* or *CCNA_00470* confer a 'rough' non-mucoid phenotype in all three backgrounds (*WT*, Δ*pleC*, and Δ*hvyA*). (**C**) Mucoidy of *Caulobacter WT* (NA1000), Δ*pleC*, or Δ*hvyA* cells over-expressing *hvyA* under control of $P_{van}$ on a medium copy number plasmid (pMT335) or $P_{xyl}$ on a low copy number plasmid (pMT375). Over-expression of *hvyA* confers the typical non-mucoid 'rough' colony phenotype on PYE agar plates supplemented with 3% sucrose, while the WT, Δ*pleC*, and Δ*hvyA* cells have a mucoid 'smooth' colony appearance. Sensitivity to bacteriophage φCr30 and buoyancy of *Caulobacter WT* (NA1000), Δ*hvyA*, Δ*pleC*, and Δ*mucR1/2* strains carrying pMT335-*hvyA* or pMT375-*hvyA* are shown in **Figure 2—figure supplement 1–2**. RsaA extracted from the same *Caulobacter* strains

*Figure 2. Continued on next page*

*Figure 2. Continued*

shown in *Figure 2* is displayed in *Figure 2—figure supplement 3*. The effect of proteinase K treatment on CCNA_00168 in *Caulobacter WT*, Δ*hvyA*, and Δ*CCNA_00163* is shown in *Figure 2—figure supplement 4*. The effect of the Δ*hvyA* mutation on swarming motility of *Caulobacter* is shown in *Figure 2—figure supplement 5*.

The following figure supplements are available for figure 2:

**Figure supplement 1**. Over-expression of *hvyA* renders Δ*pleC* cells 'heavy' and sensitive to φCr30.

**Figure supplement 2**. Over-expression of *hvyA* renders Δ*mucR1/2* cells 'heavy'. Buoyancy of Δ*mucR1/R2* cells harbouring *hvyA* on plasmid.

**Figure supplement 3**. S-layer is correctly assembled in *Caulobacter* mutants affected in capsule production.

**Figure supplement 4**. Capsule protects cell surface proteins from proteinase K treatment.

**Figure supplement 5**. Loss of *hvyA* affects swarming motility.

under the control of the vanillate inducible promoter ($P_{van}$) on a medium–copy number plasmid (pMT335 (*Thanbichler et al., 2007*)) restored the *WT* buoyancy phenotype, showing that the 'heavy' phenotype is attributable to the loss of *CCNA_00162*, *CCNA_00163*, *CCNA_00164*, *CCNA_00167*, or *CCNA_00168* function (*Supplementary file 1*).

The second cluster of genes resides on a 26-kbp mobile genetic element (MGE) that has previously been implicated in buoyancy (*Marks et al., 2010*) (*Figure 1C*, lower panel). Specifically, we recovered Tn insertions in genes predicted to encode a homolog of the putative N-acetyl-L-fucosamine transferase WbuB from *E. coli* that is involved in O-antigen (O26) synthesis (*D'Souza et al., 2002*) (*CCNA_03998*), a polysaccharide polymerase (*CCNA_00470*) and a GDP-L-fucose synthase (*CCNA_00471*). The three genes are near other coding sequences for polysaccharides biosynthesis proteins, including two other putative glycosyltransferases (*CCNA_00466* and *CCNA_00469*), a Wzx-like polysaccharide flippase/ translocase (*CCNA_00467*) and a sugar mutase homolog (*CCNA_00465*) (*Marks et al., 2010*). Consistent with a previous genome-wide Tn analysis showing that these genes cannot be disrupted (*Christen et al., 2011*), we were unable to engineer in-frame deletions in *CCNA_00466* or *CCNA_00467* in the absence of a complementing plasmid. As the entire 26-kb MGE is dispensable for viability, it appears that inactivation of either one of these four genes results in synthetic toxicity due to a polysaccharide intermediate or the sequestration of Und~P that is also required for peptidoglycan synthesis (*Yother, 2011*). To confirm that *CCNA_03998*, *CCNA_00470*, and *CCNA_00471* are indeed buoyancy determinants, we engineered in-frame deletions in *CCNA_03998* or *CCNA_00470* (*CCNA_00471* was not tested) in *WT* or Δ*pleC* mutant cells and found the resulting single or double mutants to be 'heavy' (*Figures 1C and 2A*, *Supplementary file 1*).

The association of the 'heavy' phenotype with a capsule synthesis and/or export defect, the resemblance of *CCNA_03998* to the predicted N-acetyl-L-fucosamine transferase WbuB (*D'Souza et al., 2002*) and the fact that D-fucose is a known component of the extracellular polysaccharide or capsule of *C. crescentus* (*Ravenscroft et al., 1991*) prompted us to test if these mutations also affected colony mucoidy, a phenotype typically associated with the presence of capsule or exopolysaccharides. Indeed, all 'heavy' mutants exhibited a non-mucoid ('rough') colony phenotype on PYE agar plates supplemented with 3% sucrose, while the *WT* or 'light' mutants (Δ*pleC*) had a mucoid ('smooth') colony appearance (*Figure 2A,B*, *Supplementary file 1*). In support of this result, we purified capsule from *WT Caulobacter*, Δ*CCNA_00163* ('heavy' and 'rough'), Δ*CCNA_00166* (Δ*hvyA*, 'light' and 'smooth'), and Δ*hvyA* Δ*CCNA_00163* ('heavy' and 'rough', see below) cells. As the *Caulobacter* capsule is primarily composed of neutral monosaccharides including fucose, mannose, galactose, and glucose (*Ravenscroft et al., 1991*), we used glycosyl compositional analysis as proxy to quantify capsular material from 'heavy' and 'light' strains (See 'Materials and methods'). As shown in *Table 1*, the expected sugars (determined as % of total carbohydrate weight in the preparations) were abundant in preparations from the *WT* and the 'light' mutant (Δ*hvyA*, *Table 1*), whereas those from the 'heavy' mutants (i.e. the Δ*CCNA_00163* single mutant and the Δ*hvyA* Δ*CCNA_00163* double mutant) contained far less fucose, galactose, and mannose (up to 37-fold, 23-fold, and 5.5-fold reductions, respectively, in the Δ*hvyA*

**Table 1.** Glycosyl composition of per-*O*-trimethylsilyl (TMS) derivatives of methyl glycosides performed on purified capsular polysaccharides from *WT Caulobacter* (NA1000), the single mutants Δ*CCNA_00166* (Δ*hvyA*) and Δ*CCNA_00163* and the Δ*hvyA* Δ*CCNA_00163* double mutant

| | NA1000 | | Δ*hvyA* | | Δ*CCNA_00163* | | Δ*hvyA* Δ*CCNA_00163* | |
|---|---|---|---|---|---|---|---|---|
| **Glycosyl residue** | **Mass (μg)** | **Weight (%)** | **Mass (μg)** | **Weight (%)** | **Mass (μg)** | **Weight (%)** | **Mass (μg)** | **Weight (%)** |
| Ribose | 0.8 | 0.4 | 0.1 | 0.1 | 0.3 | 0.3 | 0.5 | 0.8 |
| Rhamnose | 2.3 | 1.2 | 1.1 | 0.6 | 6.4 | 6.2 | 2.7 | 4.3 |
| Fucose | 19.8 | 10.3 | 26.5 | 14.7 | 0.0 | 0.0 | 0.2 | 0.4 |
| Xylose | 0.0 | 0.0 | 0.1 | 0.1 | 0.0 | 0.0 | 0.0 | 0.0 |
| Glucuronic Acid | 0.0 | 0.0 | 0.0 | 0.0 | 0.8 | 0.8 | 0.3 | 0.5 |
| Galacturonic acid | 28.4 | 14.9 | 32.6 | 18.1 | 5.6 | 5.4 | 2.3 | 3.7 |
| Mannose | 23.2 | 12.1 | 26.6 | 14.8 | 3.9 | 3.8 | 1.7 | 2.7 |
| Galactose | 30.1 | 15.7 | 37.0 | 20.6 | 1.4 | 1.4 | 0.5 | 0.9 |
| Glucose | 64.3 | 33.6 | 52.9 | 29.4 | 49.9 | 48.5 | 23.7 | 38.3 |
| N-Acetyl galactosamine | 2.0 | 1.1 | 0.0 | 0.0 | 2.3 | 2.2 | 1.0 | 1.7 |
| N-Acetyl glucosamine | 16.0 | 8.4 | 3.0 | 1.6 | 29.5 | 28.7 | 27.9 | 45.2 |
| N-Acetyl mannosamine | 4.1 | 2.2 | 0.0 | 0.0 | 2.2 | 2.1 | 0.7 | 1.2 |
| Σ= | 191.3 | | 179.9 | | 102.8 | | 61.8 | |

Mass is expressed in μg and weight % is relative to the total carbohydrate.

Δ*CCNA_00163* double mutant compared to the Δ*hvyA* single mutant). We also observed a significant reduction in galacturonic acid in the preparations from the 'heavy' mutants vs *WT* or 'light' cells, raising the possibility that this saccharide is also a constituent of the NA1000 capsule (*Table 1*).

Taken together, our results show that the loss of capsule synthesis and/or export renders cells 'heavy' and 'rough' and that the loss of capsulation is epistatic to the loss of PleC in buoyancy control. On these grounds we predicted that PleC, directly or indirectly, regulates one of the newly identified buoyancy determinants.

## Negative control of encapsulation by the transglutaminase homolog HvyA

Since the buoyancy switch is cell cycle-regulated and since all Δ*pleC* cells are 'light' (*Figure 1B*), we reasoned that PleC is required to turn off capsule synthesis in G1-phase SW cells. As PleC is also required to activate motility and PilA (the structural subunit of the pilus filament) expression, when Δ*pleC* divides two daughter cells that are capsulated, non-piliated and non-motile are formed. By contrast division of *WT* yields one piliated, motile, and non-capsulated ('heavy') G1-phase SW progeny and one capsulated ('light') S-phase ST cell (*Figure 1B*).

If PleC indeed restricts capsulation temporally, it might control expression of a negative regulator of capsulation. Interestingly, PleC is required for the accumulation of the transcript of *CCNA_00166* (referred to as *hvyA* due to its requirement to render cells 'heavy') (*Chen et al., 2006*). The *hvyA* transcript peaks during the G1-phase and encodes a 272-residue protein (*Figure 3A*) harbouring a classical N-terminal Sec-dependent s̲i̲gnal s̲e̲quence (SS), but lacking discernible hydrophobic sequences or a lipidation signal for retention in the membrane, suggesting that it is periplasmic. The C-terminal part of HvyA features a BTLCP domain. This domain is thought to introduce intra- or inter-molecular crosslinks by transamidation, forming γ-glutamyl-ε-lysine isopeptide bonds between Gln and Lys residues, to hydrolyse amide bonds by the reverse protease reaction and/or to execute deamidation/esterification reactions of glutamine residues (*Lorand and Graham, 2003*; *Ginalski et al., 2004*). Cysteine proteases typically feature a Cys-His-Asp catalytic triad (C192, H226, and D241 for HvyA, based on sequence alignment, *Figure 3—figure supplement 1*) for the formation of a thioester bond intermediate by the reaction of the active site thiol (from the Cys) with Gln, followed by the transfer of the acyl group to an amine substrate (from the Lys) (*Lorand and Graham, 2003*). As an in-frame deletion in *hvyA* (Δ*hvyA*) phenocopied the buoyancy defect of Δ*pleC* cells (yielding exclusively 'light' mucoid cells, *Figures 1b, 2a, 2b*), we conclude that HvyA is required

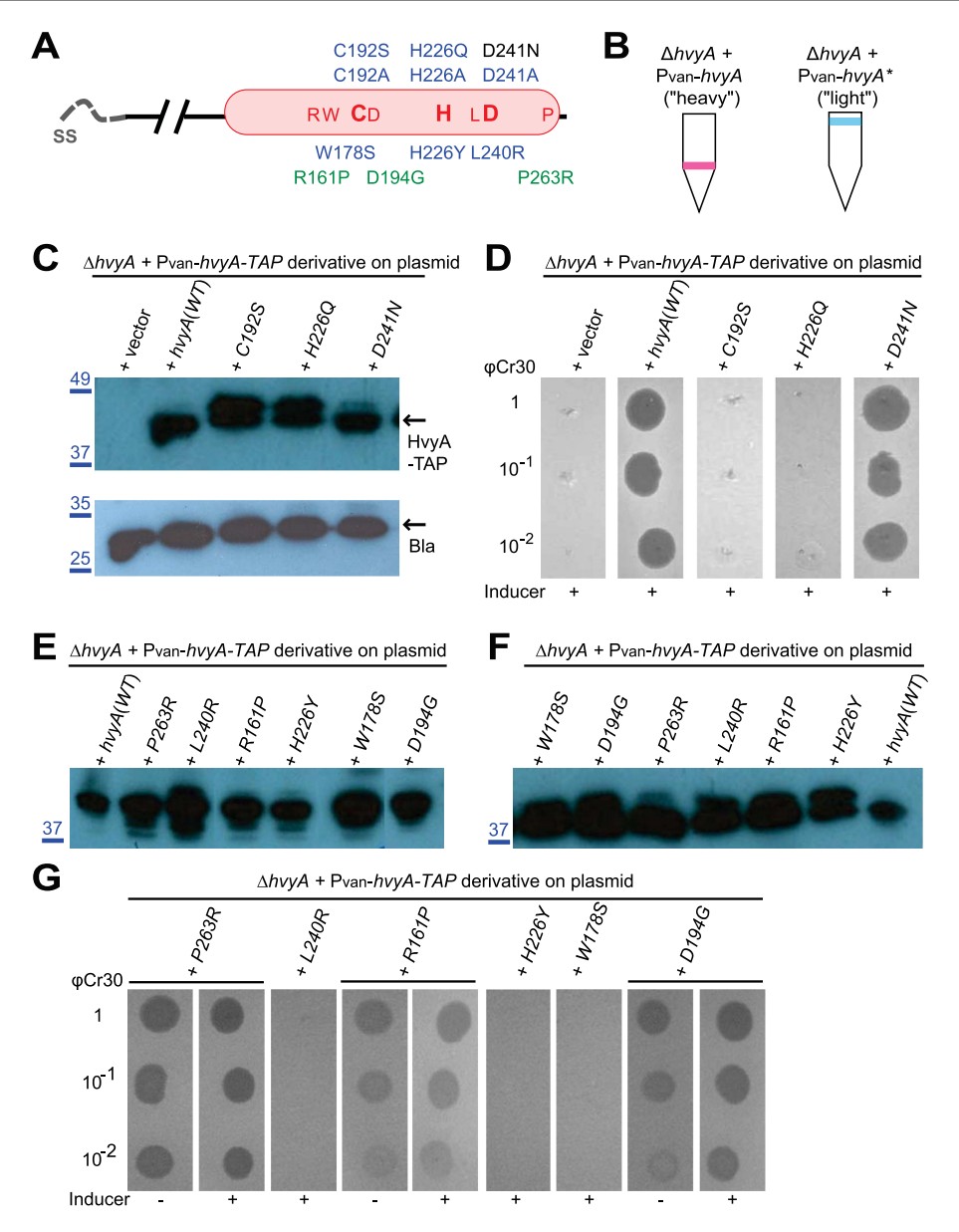

**Figure 3**. HvyA is a <u>b</u>acterial <u>t</u>ransglutaminase-<u>l</u>ike <u>c</u>ysteine <u>p</u>rotease (BTLCP) homologue and its catalytic activity is required for function. (**A**) Schematic of HvyA domains: signal sequence (SS) and BTLCP domain (in red) are indicated. C192, H226, and D241 constitute the putative catalytic triad (C192S/A, H226Q/A, and D241A alleles are non-functional; the D241N allele is functional, consistently with some BTLCP family members having a C/H/N catalytic triad). Residues identified in the buoyancy screen for non-functional variants are indicated below (blue, non-functional; green, partially functional). (**B**) Schematic of the buoyancy screen for HvyA non-functional variants. The $P_{van}$-hvyA::TAP fusion (on plasmid) was subjected to random mutagenesis, then introduced into *Caulobacter* cells that were subjected to multiple rounds of enrichment for 'light' phenotype by centrifugation on density gradient. (**C**) Immunoblot anti-HvyA-TAP on periplasmic proteins extracted by EGTA treatment. The HvyA alleles mutated in putative catalytic residues are expressed and exported to the periplasm like the WT protein. Immunoblot against *Caulobacter* β-lactamase (CCNA_02223, Bla on the lower panel) is a control for periplasmic proteins. Molecular size standards are indicated in blue on the left, with the corresponding values in kDa. (**D**) Δ*hvyA* strains harbouring *hvyA* catalytic mutants under control of $P_{van}$ on plasmid were tested for sensitivity to φCr30. Over-expression of the C192S or H226Q alleles does not restore sensitivity to φCr30, indicating that these alleles are non-functional. (**E**) Immunoblot anti-HvyA-TAP on Δ*hvyA* cells harbouring mutagenized $P_{van}$-hvyA-TAP and selected for 'light' buoyancy. The $P_{van}$-HvyA-TAP mutant alleles selected are still over-expressed. Molecular size standards are indicated in blue on the left, with

*Figure 3. Continued on next page*

*Figure 3. Continued*

the corresponding values in kDa. (**F**) Immunoblot anti-HvyA-TAP on EGTA fractions of the same clones shown in panel (**E**). The HvyA-TAP mutant alleles selected are exported to the periplasm like WT HvyA-TAP. Molecular size standards are indicated in blue on the left, with the corresponding values in kDa. (**G**) Δ*hvyA* strains harbouring *hvyA* variants under control of P*van* on pMT335 were tested for sensitivity to φCr30. The L240R, H226Y, and W178S alleles do not restore sensitivity to φCr30, whereas the P263R, R161P and D194G variants partially restore sensitivity to φCr30. The alignment of BTLCPs protein sequences from *Caulobacter* (HvyA), *S. meliloti* (SMc00998), *S. fredii* NGR234 (NGR_c12490), and *P. fluorescens* (PFL_0130) is shown in *Figure 3—figure supplement 1*. Immunoblots against HvyA-TAP, CtrA, and β-lactamase on whole lysates and EGTA fractions of cells expressing HvyA point mutants are shown in *Figure 3—figure supplement 2–3*.

The following figure supplements are available for figure 3:

**Figure supplement 1**. Conservation of the BTLCP domain in HvyA orthologs.

**Figure supplement 2**. Catalytic HvyA-TAP mutants are still exported to the periplasm.

**Figure supplement 3**. Over-expression of HvyA-TAP alleles.

for capsule-mediated buoyancy control in *Caulobacter*. While expression of *WT* HvyA from P*van* (pMT335-*hvyA*) reversed the buoyancy defect of Δ*hvyA* and Δ*pleC* cells, analogous plasmids encoding the predicted catalytic mutants (C192S/A, H226Q/A, or D241A) were unable to do so, although all the HvyA variants accumulated to comparable steady-state levels as the *WT* protein on immunoblots (*Figure 3C*, *Figure 3—figure supplement 2*). Thus expression of catalytically active HvyA is necessary and sufficient to mitigate the buoyancy defect of Δ*pleC* cells. As a genetic selection for 'heavy' Δ*hvyA*::Tn mutants was answered by Tn insertions in the same genes that render Δ*pleC* cells 'heavy' (*Figure 1C*), we reasoned that mutations in capsule synthesis and export genes are epistatic to both the Δ*hvyA* and Δ*pleC* mutations. To confirm this notion, we engineered Δ*hvyA* Δ*CCNA_00163*, Δ*hvyA* Δ*CCNA_00167*, and Δ*hvyA* Δ*CCNA_00470* double mutants as well as a Δ*pleC* Δ*hvyA* Δ*CCNA_00167* triple mutant and found all resulting mutants to be 'heavy' and non-mucoid ('rough') on PYE sucrose plates (*Figure 2A, B* and *Supplementary File 1*). Importantly, constitutive expression of HvyA (from a P*van*- or a P*xyl*-plasmid) in *WT*, Δ*pleC*, or Δ*hvyA* cells also renders cells 'heavy' and 'rough' (*Figure 2C*, *Figure 2—figure supplement 1*). On the basis of these findings we hypothesized that HvyA is a G1-specific negative regulator of capsulation whose expression is dependent on PleC (*Figure 1A*).

## Microscopic visualization of the capsule

As all previous efforts to visualize the capsule directly by light or electron microscopy had been unsuccessful (*Ravenscroft et al., 1991*), we conducted negative stain fluorescence microscopy (FM) with fluorescein isothiocyanate (FITC)-coupled dextran to measure the zone of exclusion of FITC-dextran in capsulated (Δ*hvyA*, 'light' mucoid) and non-capsulated (Δ*CCNA_00163*, 'heavy' non-mucoid) cells (*Figure 4A,B*). Akin to the difference between capsulated and non-capsulated *Streptococcus pneumoniae* (*Hathaway et al, 2012*; *Schaffner et al, 2014*), the zone of exclusion of FITC-dextran was significantly smaller in the case of Δ*CCNA_00163* compared to Δ*hvyA* cells, although the actual size of the cells by differential interference contrast (DIC) microscopy was comparable (*Figure 4A*). The increase in the exclusion radius of the dextran polymer can be explained by the presence of a capsule on Δ*hvyA* cells and by its absence from Δ*CCNA_00163* cells. Atomic force microscopy (AFM) (*Dufrêne, 2014*) provided additional support for this interpretation. In these experiments, bacteria were immobilized on porous membranes, a method allowing AFM imaging of the bacteria in liquid medium. However, recording reliable images of live cells turned out to be very difficult for two reasons: (i) most cells were detached or pushed aside during scanning; (ii) strong interactions between the tip and the soft bacterial surface led to fuzzy images that were difficult to interpret. Therefore, cells were imaged after fixation with 4% paraformaldehyde, a protocol that is known to preserve cellular structures (*Chao and Zhang, 2011*). *Figure 5A–D* shows representative low-resolution deflection images of fixed cells from the Δ*CCNA_00163* and Δ*hvyA* strains (from two independent preparations for each). As opposed to Δ*CCNA_00163* cells, which were readily imaged without apparent cell surface damage (*Figure 5A*, *Figure 5C*), the surface morphology of Δ*hvyA* cells (*Figure 5B*, *Figure 5D*) was

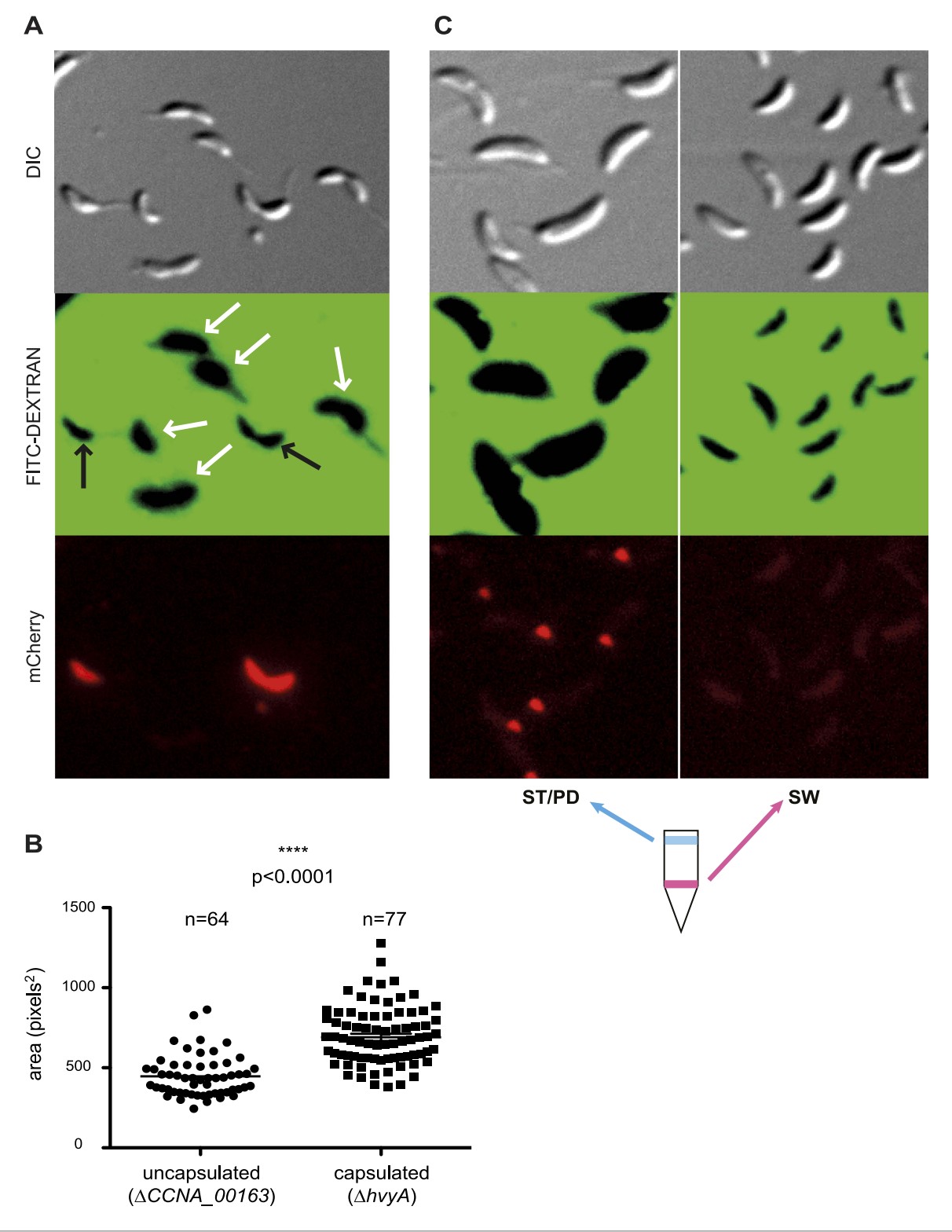

**Figure 4**. Negative fluorescence imaging of capsule. (**A**) Differential Interference Contrast (DIC, top) and fluorescence images showing FITC-dextran exclusion (middle) or mCherry (bottom) of mixed Δ*CCNA_00163* (expressing mCherry) and Δ*hvyA* cells. The area of FITC-dextran exclusion of the non-capsulated Δ*CCNA_00163* cells is much smaller than that of the capsulated Δ*hvyA* cells. (**B**) Statistical analysis of the area from which FITC-dextran was excluded. The analysis was performed as described in the 'Materials and methods'. (**C**) DIC (top) and fluorescence images showing FITC-dextran

*Figure 4. Continued on next page*

*Figure 4. Continued*

exclusion (middle) or mCherry (bottom) on NA1000 *spmX-mCherry* cells. SW and ST/PD cells were separated on density gradient before incubation with FITC-dextran. Area of exclusion of ST and PD cells ('light' cells, on the left) is much bigger than that of SW cells (on the right). SpmX-mCherry is absent from SW cells, accumulates at the SW to ST cell transition and labels the ST pole.

strongly altered by the scanning tip, and showed streaks in the scanning direction, reflecting strong interactions between the tip and the soft cell surface. Similar trends were observed when recording high-resolution images on top of individual cells (*Figure 5E–H*). While the surface of ΔCCNA_00163 cell was smooth (surface roughness on 0.06 µm² areas of ~0.9 nm) and featureless, Δ*hvyA* cells were rougher (roughness of ~2.2 nm) and showed streaks in the scanning direction, suggesting that soft, loosely bound material was pushed away by the tip. In light of earlier AFM studies (*Dague et al., 2008*), we note that such smooth and rough morphologies are consistent with the presence of crystalline-like arrays of proteins and of an amorphous layer of polysaccharides, respectively.

Taken together our results show that ΔCCNA_00163 cells are devoid of capsule, while a soft layer of capsular polysaccharides covers Δ*hvyA* cells. The results above raised the possibility that HvyA normally prevents capsulation in G1-phase cells, and FITC-dextran staining of an NA1000 culture expressing a ST cell marker (SpmX-mCherry) indeed revealed that G1-phase cells (i.e. SW cells isolated on density gradient) did not exclude the polymer and are thus non-capsulated, whereas ST/PD cells ('light' cells on density gradient) show a much bigger area of FITC-dextran exclusion (*Figure 4C*).

## HvyA abundance is restricted to G1-phase

The results above suggested that the G1-phase SW cells are able to synthesize capsule if HvyA is absent. To test if capsule export proteins are normally present in G1-phase cells, we raised antibodies to CCNA_00162, CCNA_00163, CCNA_00164, CCNA_00167, and CCNA_00168. Immunoblotting revealed that these components are present throughout the cell cycle of *WT* cells (*Figure 6A*). Owing to the poor specificity of the antibodies raised against HvyA, we engineered cells encoding a functional mCherry (mCh)-tagged HvyA derivative (mCh-HvyA) in which the mCherry moiety is fused in-frame after the SS. mCh-HvyA is expressed from the native *hvyA* promoter at the *hvyA* locus in lieu of untagged HvyA. Immunoblotting using polyclonal antibodies to mCherry revealed that mCh-HvyA is indeed restricted to G1-phase (*Figure 6A*). Antibiotic chase experiments showed that mCh-HvyA is a very unstable protein and is rapidly degraded akin to the unstable protein CtrA (*Figure 6B*), indicating that accumulation of HvyA is dictated by the timing of its synthesis (G1-phase).

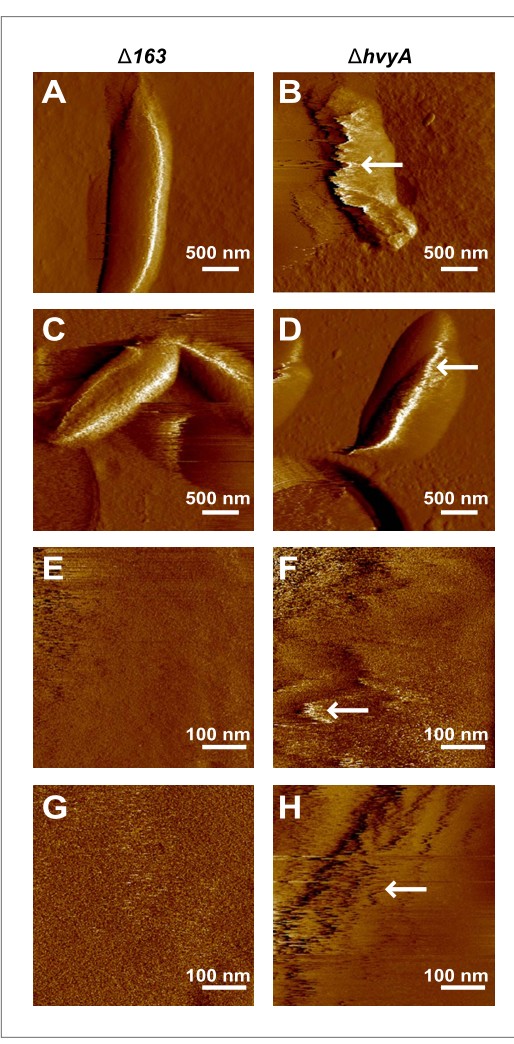

**Figure 5**. Nanoscale AFM imaging reveals the ultrastructure of *Caulobacter* cell surface. Contact-mode deflection images of ΔCCNA_00163 cells (**A**, **C**, **E**, **G**) and Δ*hvyA* cells (**B**, **D**, **F**, **H**) at low (**A–D**) and high (**E–H**) magnifications. White arrows indicate streaks generated by the AFM tip scanning the soft, loosely bound layer at the surface of Δ*hvyA* cells. Images were taken on bacteria from two independent cultures for each strain.

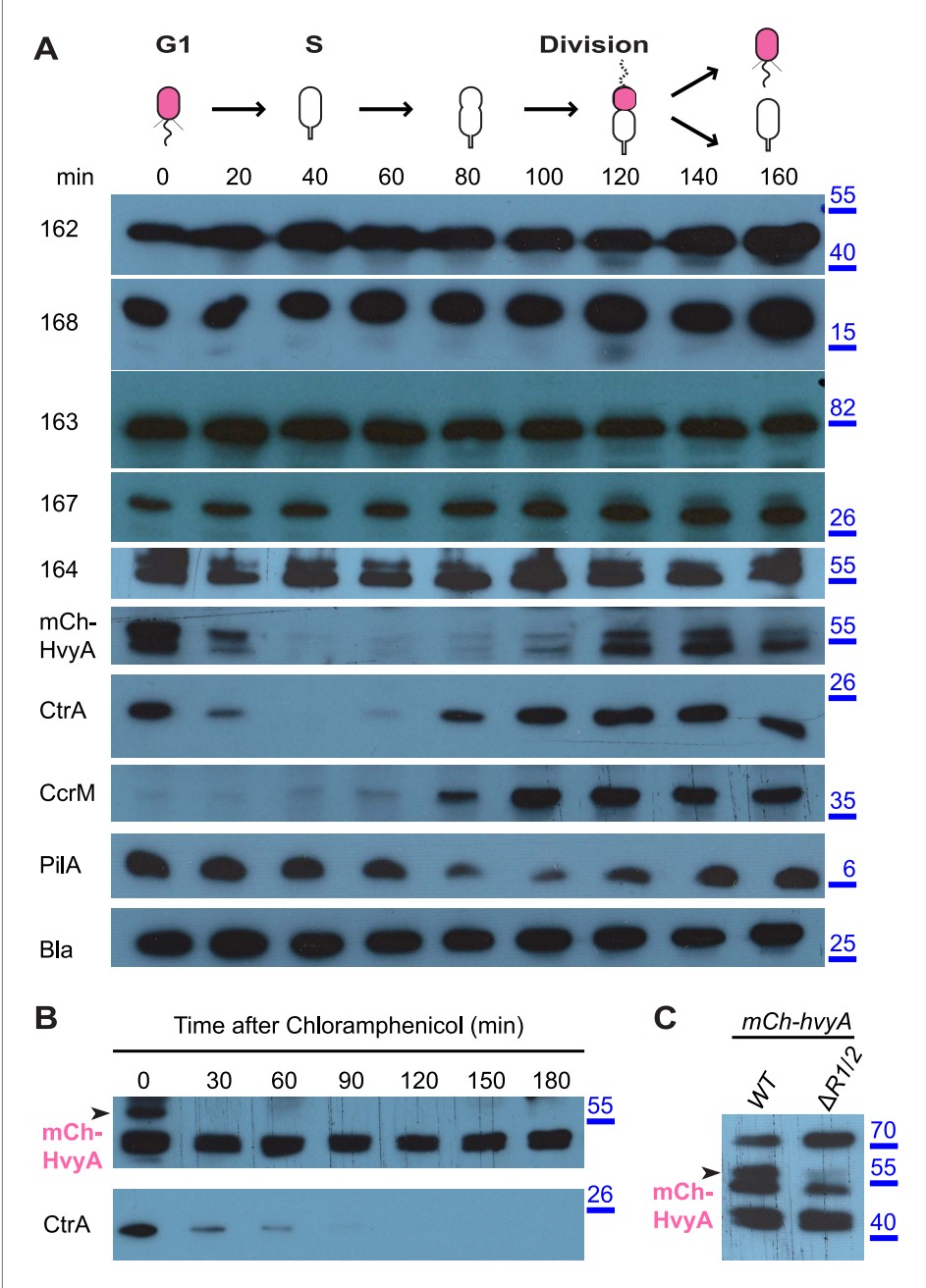

**Figure 6**. HvyA is an unstable protein and its presence is restricted to G1-phase (SW cells). (**A**) Immunoblots showing protein levels in synchronized *Caulobacter* cells: CCNA_00162, CCNA_00163, CCNA_00164, CCNA_00167, and CCNA_00168 levels do not change over the cell cycle. mCherry-HvyA accumulates only in SW cells, akin to CtrA. CcrM accumulates in PD cells, and PilA levels increase upon re-accumulation of CtrA. The β-lactamase (Bla) was used as control as it is constitutively present along the cell cycle. Molecular size standards are indicated in blue on the right, with the corresponding values in kDa. (**B**) Immunoblots showing degradation of mCherry-HvyA (indicated by the arrowhead) after addition of chloramphenicol (2 µg/ml) to stop protein synthesis. mCherry-HvyA is degraded as rapidly as the unstable protein CtrA (lower panel). Molecular size standards are indicated in blue on the right, with the corresponding values in kDa. (**C**) Immunoblot anti-mCherry-HvyA showing that levels of the mCherry-HvyA fusion protein (indicated by the arrowhead) are significantly decreased in the Δ*mucR1/2* mutant strain. Molecular size standards are indicated in blue on the right, with the corresponding values in kDa. Levels of CCNA_00162, CCNA_00163, CCNA_00164, CCNA_00167, and CCNA_00168 in *Caulobacter* cells over-expressing HvyA (from P$_{van}$) are shown in **Figure 6—figure supplement 1**.

*Figure 6. Continued on next page*

*Figure 6. Continued*

The following figure supplement is available for figure 6:

**Figure supplement 1**. Over-expression of *hvyA* does not alter the levels of capsule polymerization/export components.

---

Next, we explored if the G1-specific regulation of HvyA is due to transcriptional regulation by PleC. Using a *lacZ*-based promoter-probe reporter in which the *hvyA* promoter is transcriptionally fused to the *lacZ* gene (P$_{hvyA}$-*lacZ*), we found that P$_{hvyA}$ is indeed positively dependent on PleC (41.1 ± 6.0% of *WT* activity, *Figure 7A*). Inactivation of the gene encoding the antagonistic DivJ kinase in Δ*pleC* cells mitigated this response (94.0 ± 19.9% of *WT* activity, *Figure 7A*). Consistent with the fact that PleC and DivJ regulate CtrA~P levels, our recent chromatin immunoprecipitation coupled to deep-sequencing (ChIP-Seq) analyses indicated that CtrA indeed binds the *hvyA* promoter (P$_{hvyA}$) and that this binding is strongly diminished in Δ*pleC* cells (*Figure 7—figure supplement 1*). Moreover, P$_{hvyA}$-*lacZ* is poorly active in *ctrA401* mutant cells (43.3 ± 1.9% of *WT* activity, *Figure 7A*) that express a temperature-sensitive version of CtrA (T170I) in lieu of *WT* CtrA (*Quon et al., 1996*). By contrast, P$_{hvyA}$-*lacZ* is strongly de-repressed (574 ± 30% of *WT* activity, *Figure 7A*) in the absence of the paralogous repressors MucR1 and MucR2 (Δ*mucR1/2*) that directly bind P$_{hvyA}$ (*Figure 7A*, *Figure 7—figure supplement 1*) and that silence many promoters of G1-phase genes (i.e. those that are activated by CtrA~P after compartmentalization (*Fumeaux et al., 2014*)). Thus, CtrA~P and MucR1/2 directly activate and repress P$_{hvyA}$, respectively (see *Figure 8*). Consistent with the latter, MucR1/2 also represses P$_{hvyA}$-*lacZ* activity in Δ*pleC* cells over-expressing *WT* or a phosphomimetic variant (D51E) of CtrA (*Figure 7—figure supplement 2*). Thus, *hvyA* (and other genes whose transcripts peak in G1, such as *pilA* and *sciP*) is expressed from a promoter that is temporally confined during the cell cycle via activation and repression by PleC/CtrA~P and MucR1/2, respectively.

Remarkably, regulation of P$_{hvyA}$ by MucR is conserved in *S. meliloti*. We found that P$_{hvyA}$-*lacZ* was strongly de-repressed (1683 ± 130% of *WT* activity) in a *mucR*::Tn mutant derivative (Rm101) of the *S. meliloti WT* strain (Rm2011, *Figure 9A*). Conversely, promoter probe assays in *WT* and *mucR* mutant cells of *Caulobacter* and *S. meliloti* revealed that the promoter of *SMc00998*, the *S. meliloti hvyA* ortholog that can partially substitute for *C. crescentus hvyA* (see below), is regulated by MucR in both these alpha-proteobacteria. The promoter probe plasmid P$_{SMc00998}$-*lacZ* indicated that the *SMc00998* promoter is strongly de-repressed in *S. meliloti mucR*::Tn (314 ± 25% of *WT* activity) and the *Caulobacter* Δ*mucR1/2* mutant (568 ± 25% of *WT* activity, *Figure 9B*) compared to *WT*. Recent microarray data showed that the *SMc00998* transcript peaks in G1-phase and that its promoter harbours a CtrA-binding site (*De Nisco et al., 2014*). Moreover, our ChIP-Seq analysis indicates that CtrA associates with the promoter of the *hvyA*-like gene *NGR_c12490* of *Sinorhizobium fredii* NGR234 (*Fumeaux et al., 2014*). Thus, positive and negative transcriptional regulation by CtrA and MucR is potentially wide-spread in alpha-proteobacteria.

## Cell cycle control of HvyA synthesis at the level of translation

Buoyancy tests with Δ*mucR1/2* cells unearthed another mechanism of cell cycle-control of HvyA synthesis: translational regulation. We uncovered this mechanism upon our erroneous prediction that Δ*mucR1/2* cells should be 'heavy' (non-capsulated). This prediction was based on the strong de-repression of P$_{hvyA}$ in the absence of MucR1/2 observed above and our result that overexpression of HvyA renders *Caulobacter* cells 'heavy'. To our surprise, we found that Δ*mucR1/2* cells are in fact 'light' (*Figure 2—figure supplement 2*) and only express trace amounts of mCh-HvyA (*Figure 6C*). In support of the notion that a deficiency in HvyA underlies the buoyancy defect of this strain, the activity of a P$_{hvyA}$-*hvyA::lacZ* translational fusion is indeed reduced in Δ*mucR1/2* cells (19.5 ± 5.4% of *WT* activity, *Figure 7B*), despite the massive de-repression of P$_{hvyA}$. Moreover, constitutive expression of HvyA (on plasmid, from P$_{van}$ or P$_{xyl}$) induced the 'heavy' (non-capsulated) phenotype in Δ*mucR1/2* cells (*Figure 2—figure supplement 2*). Expression of *WT* MucR1 from pMT335 alleviated both the buoyancy defect and P$_{hvyA}$ de-repression of Δ*mucR1/2* cells, while analogous plasmids encoding the mutant MucR1 derivatives Y97C or R85C were unable to do so (*Figure 7A*, *Figure 7—figure supplement 3*), indicating that these defects are indeed due to aberrant expression of a MucR1/2-dependent target gene.

Remarkably, MucRs from alpha-proteobacterial lineages can control transcription and translation of *hvyA*. In fact MucR orthologs from *Agrobacterium tumefaciens*, *Brucella suis*, *Bartonella henselae*, or

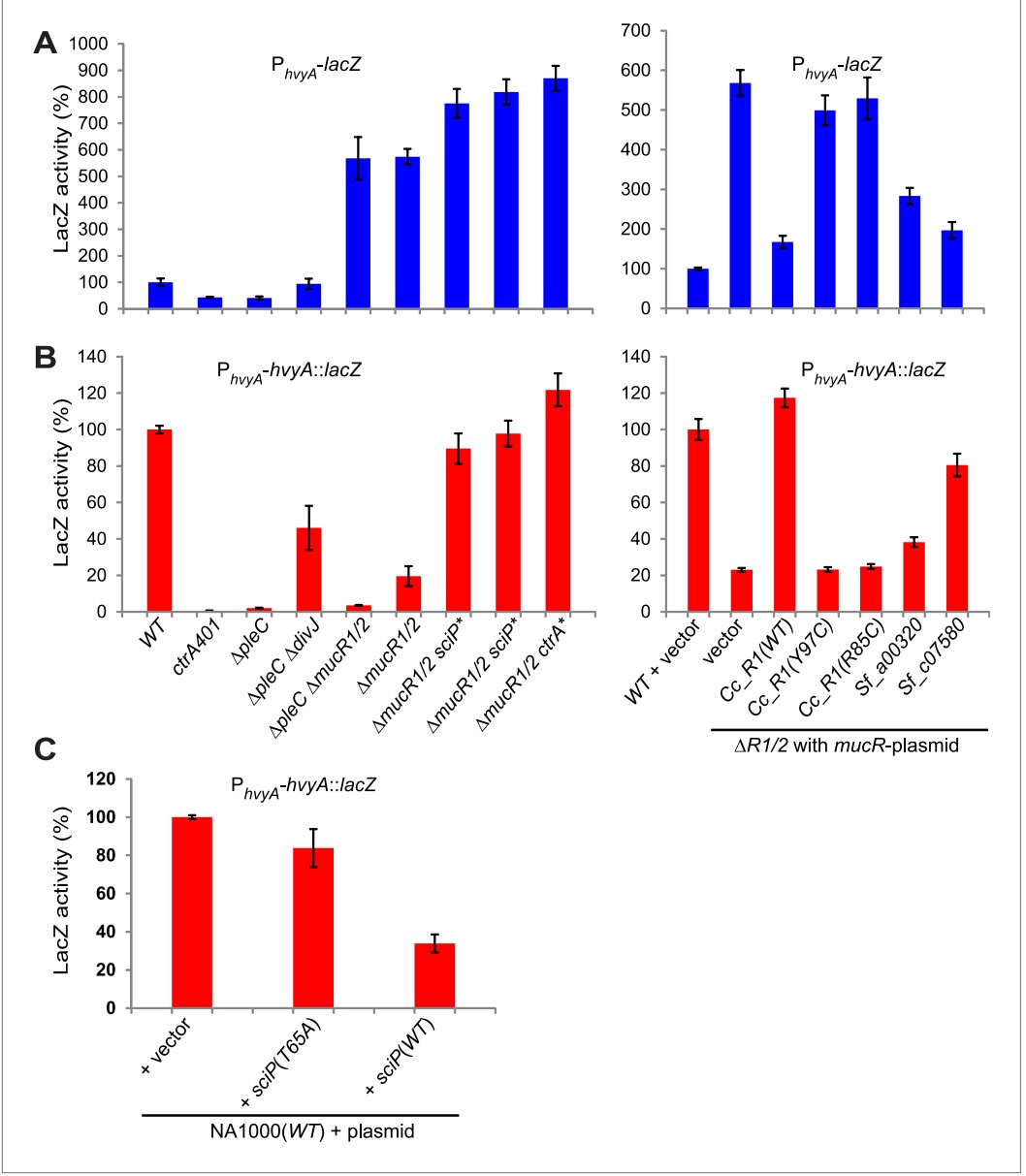

**Figure 7**. Transcriptional and translational regulation of *hvyA* depends on PleC-CtrA, SciP, and MucR1/2.
(**A**) Beta-galactosidase activity of P$_{hvyA}$-*lacZ* transcriptional fusion. Left panel: transcription of *hvyA* is strongly reduced in *ctrA401* (T170I, temperature sensitive) and Δ*pleC* strains compared to *WT Caulobacter*. Mutation of the kinase DivJ partially restores *hvyA* transcription in the Δ*pleC* Δ*divJ* strain. Transcription from P$_{hvyA}$-*lacZ* is significantly increased in the Δ*mucR1/2* mutant. Right panel: beta-galactosidase activity of P$_{hvyA}$-*lacZ* in Δ*mucR1/2* cells complemented with *Caulobacter mucR1* (*CC_R1*, *WT* or mutant derivatives Y97C and R85C) or the MucR paralogs from *S. fredii* NGR234 (*Sf_a00320* and *Sf_c07580*) on plasmid under control of P$_{van}$. Values are expressed as percentages (activity in *WT* NA1000 or *WT* carrying the empty vector set at 100%). (**B**) Beta-galactosidase activity of P$_{hvyA}$-*hvyA::lacZ* translational fusion in the same strains shown in panel (**A**). Translation of *hvyA* is strongly reduced in cells expressing the *ctrA401* allele and in the Δ*pleC* strain compared to *WT Caulobacter*. Mutation of DivJ partially restores *hvyA* translation in the Δ*pleC* Δ*divJ* strain. P$_{hvyA}$-*hvyA::lacZ* activity is also significantly decreased in the Δ*mucR1/2* and Δ*pleC* Δ*mucR1/2* mutants, consistently with the 'light' buoyancy of these strains. P$_{hvyA}$-*hvyA::lacZ* activity is restored in the Δ*mucR1/2* double mutant carrying *ctrA*(T170A), *sciP*(T24I), or *sciP*(T65A) alleles (*ctrA** and *sciP**). Values are expressed as percentages (activity in *WT* NA1000 or *WT* carrying the empty vector set at 100%). (**C**) Translational control of *hvyA* by CtrA and SciP. Beta-galactosidase activity of the P$_{hvyA}$-*hvyA::lacZ* fusion in *WT Caulobacter* cells over-expressing *sciP*(T65A) or *sciP*(WT) from P$_{van}$ on plasmid. Whereas *sciP*(T65A)
*Figure 7. Continued on next page*

*Figure 7. Continued*

does not affect the activity of the translational P$_{hvyA}$-*hvyA::lacZ* fusion, over-expression of *sciP(WT)* significantly decreases the activity of the fusion, consistently with the model presented in *Figure 8*. The activity is expressed as percentage of the activity in *WT* cells carrying the empty vector. Occupancy of CtrA, MucR1, and MucR2 at the *hvyA* promoter region as determined by ChIP-seq analysis is shown in *Figure 7—figure supplement 1*. The effect of the CtrA(D51E) allele on P$_{hvyA}$-*lacZ* is shown in *Figure 7—figure supplement 2*. The ability of heterologous MucR to restore P$_{hvyA}$-*lacZ* repression or P$_{hvyA}$-*hvyA::lacZ* activity in *Caulobacter ΔmucR1/2* is reported in *Figure 7—figure supplement 3*.

The following figure supplements are available for figure 7:

**Figure supplement 1**. Occupancy at the *hvyA* promoter region as determined by ChIP-seq analysis.

**Figure supplement 2**. Transcriptional control of *hvyA* by PleC-CtrA.

**Figure supplement 3**. Transcriptional and translational control of *hvyA* by heterologous MucRs.

*S. fredii* NGR234 (NGR_a00320 or NGR_c07580) can partially complement the *Caulobacter ΔmucR1/2* mutant. Interestingly, in the presence of a heterologous MucR, increased translational activity of P$_{hvyA}$-*hvyA::lacZ* was always accompanied with a commensurate repression of the P$_{hvyA}$-*lacZ* transcriptional reporter (*Figure 7A, B*, *Figure 7—figure supplement 3*). Specifically, while *S. fredii* NGR_c07580 and *A. tumefaciens* ROS were functional in transcriptional and translational regulation of *hvyA*, *B. henselae* MucR was inactive. However, *B. suis* MucR and *S. fredii* NGR_a00320 displayed an intermediate level of activity (47.0 ± 3.0% and 38.2 ± 2.7% of *WT* activity for P$_{hvyA}$-*hvyA::lacZ* and 302.0 ± 11.1% and 283.5 ± 20.6% of *WT* activity for P$_{hvyA}$-*lacZ*, respectively; *Figure 7A,B*, *Figure 7—figure supplement 3*).

How might MucR regulate the translation of *hvyA*? The results above, along with our finding that MucR1/2 acts pleiotropically, prompted us to speculate that another target of MucR1/2 is responsible for translational regulation of *hvyA*. Our experiments showed that the translational regulator of *hvyA* (X in *Figure 8*) is an indirect target of MucR1/2 and is deeply integrated into the cell cycle circuitry. Indeed, we observed that mutations in the master regulator gene *ctrA* [*ctrA*(T170A)] or its antagonist gene *sciP* [*sciP*(T24I) or *sciP*(T65A)] (*Fumeaux et al., 2014*) are epistatic to the *ΔmucR1/2* mutation, as they conferred near normal P$_{hvyA}$-*hvyA::lacZ* translation and a normal cellular buoyancy (capsulation) pattern with 'heavy' and 'light' cells, while P$_{hvyA}$ was still de-repressed (*Figure 7A,B*). In support of this view, over-expression of *WT* SciP from pMT335 cripples P$_{hvyA}$-*hvyA::lacZ* translation (33.9 ± 4.7% of *WT* activity), while over-expression of SciP(T65A) only has negligible effects (83.8 ± 9.9% of *WT* activity; *Figure 7C*).

Importantly, we identified mechanisms for both transcriptional and translational regulation of *hvyA* in *Caulobacter* cells and showed that these two functions can be genetically uncoupled. This regulatory complexity highlights the requirement of proper buoyancy and capsulation control during the cell cycle. Since SciP is the negative regulator of S-phase genes (activated by CtrA at the PD cell stage before compartmentalization), while MucR1/2 negatively regulate G1-specific genes (activated by CtrA after compartmentalization), the coordinated transcriptional and translational control of *hvyA* by CtrA, MucR1/2, and SciP suggests that cells prepare themselves for the impending capsule-less SW cell phase by setting the stage for rapid translation of HvyA once the transcript is synthesized at compartmentalization.

## Capsule protects from infection by the S-layer-specific bacteriophage φCr30

As it has been suggested that the MGE and the buoyancy phenotype can influence the resistance profile to the S-layer specific Caulophage φCr30 (*Edwards and Smith, 1991*; *Awarm and Smit, 1998*), we evaluated this link and the possible involvement of *hvyA* quantitatively by genome-wide Tn mutagenesis followed by deep-sequencing (Tn-Seq) of *Caulobacter* cells challenged or not with bacteriophage φCr30. This analysis revealed that *hvyA* is one of the two major φCr30-resistance determinants, along with the *rsaA* locus encoding the RsaA subunit of the S-layer and CCNA_01057 that is predicted to function in S-layer assembly (*Awram and Smit, 1998*) (*Figure 10A*). In fact, Tn insertions in *hvyA* and the *rsaA* gene are >380 and >160 times over-represented vs other genes in φCr30-challenged cells

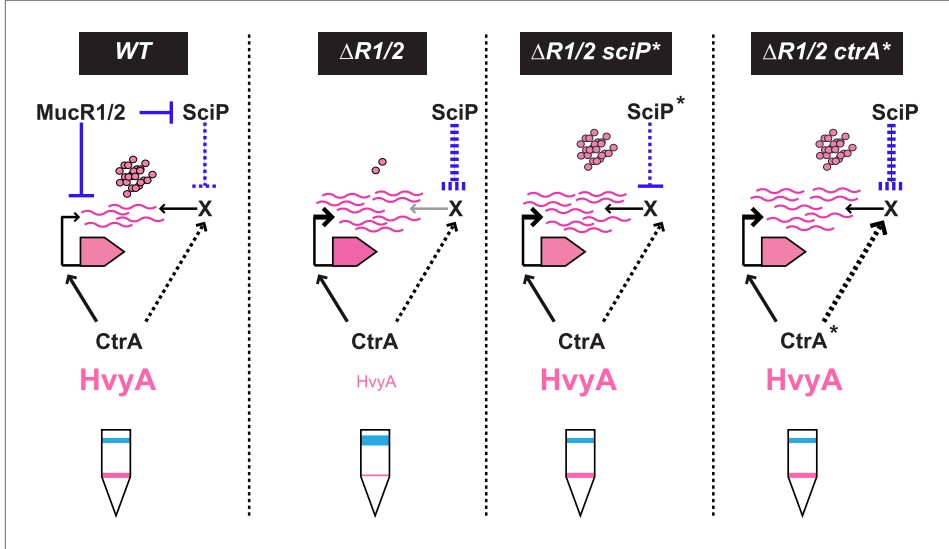

**Figure 8**. Model for regulation of HvyA synthesis and control of cell buoyancy/capsulation in *Caulobacter*. In WT cells, *hvyA* transcription is activated by CtrA and repressed by MucR1/2, whereas translation is promoted by a (still unknown) factor (X, itself under control of CtrA and SciP like many late S phase genes). This tight regulation restricts HvyA synthesis to G1-phase, where HvyA prevents the encapsulation of the SW cell. In a Δ*mucR1/2* mutant, SciP is over-produced, which lowers the levels of the translational regulator of *hvyA* and prevents HvyA protein accumulation, despite de-repression of *hvyA* transcription. The presence of the *ctrA*(T170A), *sciP*(T24I), or *sciP*(T65A) alleles (*ctrA\** or *sciP\**) restores the synthesis of the factor X in the Δ*mucR1/2* mutant background, and therefore restores HvyA synthesis and WT buoyancy.

compared to those without phage, while insertions in *CCNA_01057* are over-represented >400-fold upon phage treatment of cells, consistent with the S-layer serving as φCr30-receptor (*Edwards and Smit, 1991*). As expected we observed that insertions in the PleC-regulated *hvyA* gene confer a growth advantage in the presence of φCr30, but we also found Tn insertions in *pleC* itself to be enriched (>90-fold, *Figure 10A*). The reduction of Tn insertion bias in *pleC* compared to *hvyA* may be related to the residual expression of *hvyA* in *pleC* mutant cells (and thus lower fitness towards φCr30) compared to the complete absence of HvyA due to Tn insertions in *hvyA*.

In contrast to the positive effects of the insertions in *hvyA* upon φCr30 challenge, Tn insertions in *CCNA_00162-00168* and *CCNA_00460-00481* were strongly under-represented (>220- and >120-fold, respectively) indicating that these genes have the opposite fitness effect, as would be predicted from our epistasis experiments on capsulation (*Figure 10—figure supplement 1*). To confirm these Tn-Seq results, we conducted phage spot tests with serial dilutions of φCr30 on lawns of either *WT*, Δ*hvyA*, or Δ*pleC* cells and observed that deletion in *hvyA* or *pleC* did not yield cleared zones (cell lysis) where the phage had been spotted (*Figure 2A*). Complementation of the Δ*hvyA* strain with *WT hvyA* restored φCr30 sensitivity, whereas the catalytic point mutants (encoding C192S and H226Q) were unable to do so (*Figure 3D*). In addition, Δ*hvyA* or Δ*pleC* strains harbouring Δ*CCNA_00163 or* Δ*CCNA_00470* mutations exhibited clearing zones (lysis) akin to the *WT* (*Figure 2A*). Thus, a Δ*hvyA* or Δ*pleC* mutation renders the entire population 'light', capsulated, and resistant to φCr30, while either a Δ*CCNA_00163* or Δ*CCNA_00470* (akin to Δ*CCNA_0162*, Δ*CCNA_00164*, Δ*CCNA_00167*, *CCNA_00168::Tn* or Δ*CCNA_03998*; *Figure 2A*, *Supplementary file 1*) mutation mitigates this effect, rendering the double mutants 'heavy', non-capsulated, and sensitive to φCr30. Moreover, the fact that mutations in *pleC*, *hvyA*, or *CCNA_00163* did not noticeably affect the release of the RsaA S-layer subunit from the cell surface (*Figure 2—figure supplement 3A*) indicates that the S-layer is present and properly assembled in all the mutants.

On the basis of these results, we hypothesize that the capsule protects the S-layer from φCr30 (as depicted in the model in *Figure 10B*). Consistent with a protective role of the capsule for the cell surface and our results in spot tests with φCr30, we also observed higher frequencies of generalized transduction by φCr30 for non-capsulated strains compared to *WT*, Δ*pleC*, or Δ*hvyA* strains (*Table 2*).

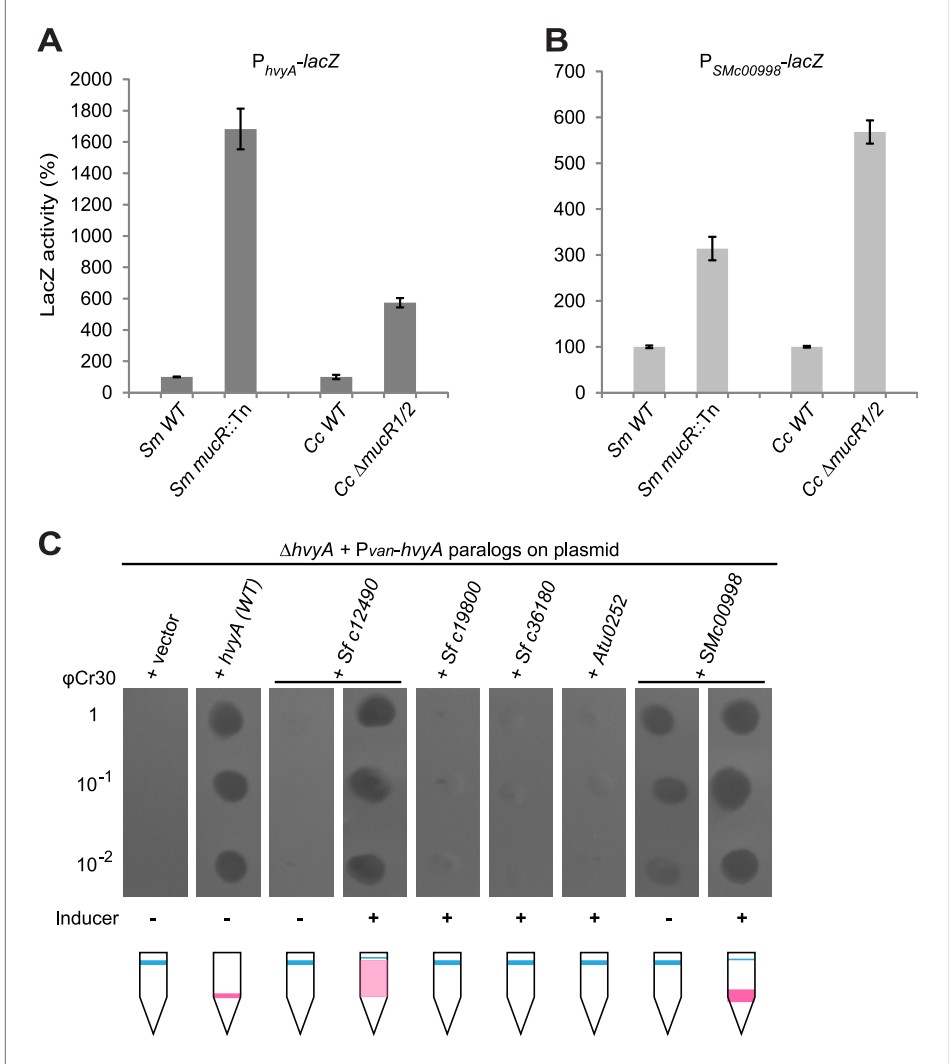

**Figure 9**. Conservation of *hvyA* transcriptional control and BTLCP function in alpha-proteobacteria. (**A**) Beta-galactosidase activity of the P*hvyA*-*lacZ* transcriptional fusion in *WT* and Δ*mucR1/2* cells compared to *S. meliloti* *WT* (Rm2011) and *mucR* mutant (*mucR*::Tn, Rm101). The assay shows that the transcriptional repression of P*hvyA*-*lacZ* by MucR is conserved in *S. meliloti*. The activity is expressed as the percentage of the activity in (*Caulobacter* or *S. meliloti*) *WT* cells. (**B**) Beta-galactosidase activity of the P*SMc00998*-*lacZ* transcriptional fusion in *WT* and Δ*mucR1/2* cells compared to *S. meliloti* *WT* (Rm2011) and *mucR* mutant (*mucR*::Tn, Rm101). The assay shows that the HvyA paralog SMc00998 is also under transcriptional repression by MucR in both *S. meliloti* and *Caulobacter*. The activity is expressed as the percentage of the activity in (*Caulobacter* or *S. meliloti*) *WT* cells. (**C**) Sensitivity to φCr30 and buoyancy of *C. crescentus* Δ*hvyA* cells harbouring HvyA paralogs under control of P*van* on plasmid. Over-expression of *NGR_c12490* or *SMc00998* restores sensitivity to φCr30 and (partially) cell buoyancy, whereas *NGR_c19800*, *NGR_c36180*, or the *A. tumefaciens* paralog (*Atu0252*) are unable to complement the Δ*hvyA* mutation.

At the same time, treatment of intact cells with proteinase K showed that outer membrane proteins such as CCNA_00168 are more sensitive to the protease in Δ*CCNA_00163* mutant cells compared to *WT* or Δ*hvyA* strain (*Figure 2—figure supplement 4*), which also supports a protective role of the capsule for the cell surface and illustrates its role in preventing φCr30 infection in *Caulobacter*.

## Conservation of HvyA function in other alpha-proteobacteria

The connection among capsulation, buoyancy, and φCr30 resistance phenotypes prompted us to probe these phenotypes in structure–function analyses with HvyA. To this end, we exploited the HvyA

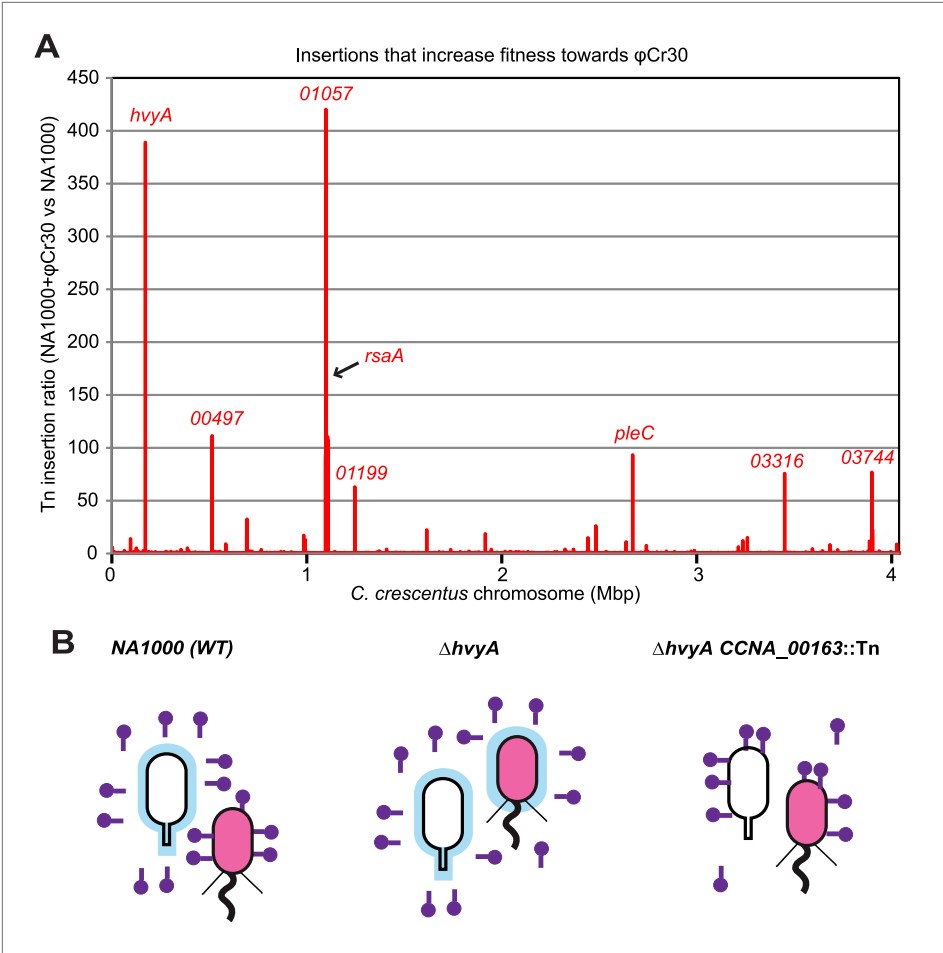

**Figure 10**. Inactivation of HvyA and ectopic presence of capsule protect cells from phage infection. (**A**) Tn insertion bias in coding sequences (CDS) of NA1000 + φCr30 relative to *WT* cells as determined by Tn-seq. Peaks show CDSs with the highest number of Tn insertions. *CCNA_01057* is part of the S-layer locus and is required for S-layer assembly. CDSs with an insertion ratio higher than 50 are indicated (*CCNA_00497*, putative rhamnosyl transferase; *CCNA_01199*, putative glucose-1-P thymidylyltransferase; *CCNA_03316*, putative UDP-N-acetylglucosamine-4,6-dehydratase; *CCNA_03744*, putative dTDP-glucose-4,6-dehydratase). Non-coding sequences are not included. (**B**) Model showing how capsule can interfere with φCr30 infection. In *WT* NA1000, φCr30 can infect SW cells, whereas ST cells are protected by the capsule (in blue) that masks the S-layer (φCr30 receptor). In a Δ*hvyA* mutant, both cell types (SW and ST) are capsulated, which significantly reduces the ability of φCr30 to infect these cells. Conversely, in a capsule-less mutant (for example Δ*hvyA CCNA_00163*::Tn, like the mutants isolated in our buoyancy screen for 'heavy' cells), both cells types (SW and ST) can be infected by φCr30. Tn insertion bias in coding sequences (CDS) of NA1000 cells relative to NA1000 + φCr30 as determined by Tn-seq is shown in *Figure 10—figure supplement 1*.

The following source data and figure supplement is available for figure 10:

**Source data 1**. (xlsx) contains the insertion ratios obtained for the Tn-Seq experiment (column **F** was used to create *Figure 10A* and column G *Figure 10—figure supplement 1*).

**Figure supplement 1**. Inactivation of capsule synthesis/export genes decreases *Caulobacter* fitness towards φCr30.

---

overexpression phenotype to seek HvyA mutant derivatives with impaired function. Briefly, a plasmid (pUG52) expressing a HvyA-derivative with a C-terminal TAP (<u>t</u>andem <u>a</u>ffinity <u>p</u>urification) tag (***Puig and et al., 2001***) under the control of P$_{van}$ was subjected to random mutagenesis and subsequently introduced into the *WT* and the Δ*hvyA* mutant strain. Whereas this plasmid normally renders cells 'heavy' (non-capsulated) due to overexpression of HvyA-TAP, serial enrichment of the mutant pool by density gradient centrifugation led to recovery of 'light' mutants (***Figure 3B***). Following immunoblotting of

**Table 2.** Transduction frequencies of φCr30 lysates in different *Caulobacter* strains

| Strain | hvyA::pSA480 | CCNA_01524::Tn |
|---|---|---|
| NA1000 (*WT*) | 344 | ~750 |
| Δ*pleC* | 64 | 165 |
| Δ*hvyA* | 48 | 162 |
| Δ*CCNA_00163* | 522 | ≥1000 |
| Δ*pleC* Δ*CCNA_00163* | 527 | ≥1000 |
| Δ*hvyA* Δ*CCNA_00163* | 487 | ≥1000 |

For transduction, cells were normalised according to the OD600 and infected with the same amount of φCr30; two different markers were transduced, a pGS18T derivative integrated at the *hvyA* locus (pSA480) and a *himar1* insertion in *CCNA_01524* (flagellar modification gene, unrelated to cellular buoyancy or capsule production). The number of colonies counted after 3 days of incubation at 30°C is reported. The experiment was repeated twice.

candidate strains for those still overexpressing HvyA-TAP (*Figure 3E*, *Figure 3—figure supplement 3A*), we recovered a number of mutants with single amino acid substitutions within the BTLCP domain of HvyA. Among these variants we recovered mutants in or very close to the putative catalytic site (H226Y, D194G, L240R), as well as mutations in conserved (W178S), similar (R161P) or non-conserved (P263R) residues throughout the BTLCP domain (*Figure 3A*, *Figure 3—figure supplement 1*). While the isolation of the mutations in the catalytic centre validated the forward genetic screen, upon re-testing the mutants three of them (R161P, D194G, and P263R) exhibited complete loss-of-function with respect to buoyancy but only partial loss-of-function with respect to φCr30 sensitivity under conditions of maximal induction of the P$_{van}$ promoter on the plasmid. In fact, these mutants were able to support replication of φCr30 (as shown by cells lysis in *Figure 3G*), indicating that they are indeed partially functional and that the φCr30 spot assay is much more sensitive as readout, possibly because localized breaches in the capsule are sufficient to promote φCr30 infection at a high multiplicity of infection (m.o.i.). By contrast, the other missense mutants from the screen (W178S, H226Y, and L240R) neither restored φCr30-sensitivity nor buoyancy (*Figure 3G*). To confirm that mutations in the BTLCP do not affect sorting to the periplasm, we released periplasmic components by treating cells with EGTA (ethylene-glycol-tetraacetic acid), which is known to destabilize the outer membrane (*Raetz et al., 2007*). Such treatment effectively liberated the beta-lactamase encoded by *Caulobacter* (CCNA_02223) along with WT or mutant HvyA-TAP variants, but not cytoplasmic proteins such as CtrA (*Figure 3C,F*; *Figure 3—figure supplement 2*, *Figure 3—figure supplement 3B*).

Prompted by the identification of the functional site in the *Caulobacter* BTLCP, we asked if heterologous BTLCPs support HvyA function in *Caulobacter*. While this was not the case for the BTLCP from *A. tumefaciens* (Atu0252) or two HvyA paralogs from *S. fredii* NGR234 (NGR_c19800 and NGR_c36180), the BTLCP homologs SMc00998 and NGR_c12490 from *S. meliloti* and *S. fredii* NGR234, respectively, restored φCr30-sensitivity and partially compensated for the buoyancy defect of Δ*hvyA* cells (*Figure 9C*). Thus, despite adaptation to dissimilar ecological niches during evolution, alpha-proteobacterial lineages have retained and exhibit related BTLCP activity.

## Discussion

The intricate transcriptional and translational regulatory circuit acting on the unstable BTLCP homolog HvyA confines its presence, and thus activity, to a specific phase (G1) in the *Caulobacter* cell cycle. An inhibitor of encapsulation, HvyA ensures that G1 cells are capsule-less and that upon its elimination during the G1→S transition encapsulation can commence. In light of this first demonstration of cell cycle-regulated capsulation and the importance of capsule in masking, preventing, or impeding immune responses in bacterial pathogens, our results raise the intriguing possibility of targeting specifically the G1-phase regulatory hierarchy to combat bacterial infections by crippling encapsulation mechanisms. This can be achieved either by inhibiting capsular functions such as export or biosynthesis or, alternatively, by potentiating or sustaining the action of capsule inhibitors, such as *Caulobacter* HvyA, throughout the cell cycle. Our genetic dissection of the regulatory complexity underlying HvyA synthesis, involving cell cycle-controlled transcription and translation, provides a possible entry point for achieving such dysregulation to prevent capsulation. Noteworthy are our results showing that the mechanisms directing *hvyA* transcription in G1-phase are also operational in other alpha-proteobacteria. Interference with HvyA proteolysis at the G1→S transition presents another strategy to obtain capsule-less cells. However, possible strategies in doing so await the identification of the periplasmic protease responsible for the proteolytic degradation of HvyA.

In addition to offering protection from immune cells, capsulation is a known resistance mechanism to bacteriophage adsorption (*Hyman and Abedon, 2010*). Our data show that in *Caulobacter* the T4-like phage φCr30 is obstructed by capsulation. As the G1-specific inhibition of capsulation by HvyA is released with the removal of HvyA during the G1→S transition, the nascent S-phase cells are protected from infection by φCr30. As φCr30 is a generalized transducing bacteriophage, this temporal confinement of capsulation also has the consequence of limiting φCr30-mediated horizontal exchange of genetic material to SW cells.

Temporal control of capsulation may also serve to prevent steric interference on the flagellar rotation by the capsule in the motile G1-phase. Steric hindrance of flagellar function by extracellular polysaccharides is well documented in other bacteria (*Blair et al., 2008*; *Jarrell and McBride, 2008*) and, consistent with this notion, we observed a slight reduction in the motility of Δ*hvyA* cells compared to WT or the Δ*hvyA* Δ*CCNA_00167* double mutant (*Figure 2—figure supplement 5*). In S-phase cells the capsule may confer protection from certain bacteriophages or other predators while also endowing S-phase cells with the buoyancy that maintains them near high oxygen tensions at the surface of aqueous environments. A strict aerobe, *Caulobacter*, and S-phase cells in particular may have demand for energy production by respiration.

BTLCPs are particularly widespread in alpha-proteobacterial genomes and are also found in certain gamma-proteobacterial clades such as *Pseudomonads* and *Vibrios*, but they appear to act on specific targets in these lineages. In *Caulobacter*, the HvyA coding sequence is embedded in the CPS export locus, but in other alpha-proteobacteria HvyA-like BTLCPs appear to be encoded in other genomic context(s), so they probably affect also phenotypes unrelated to surface polysaccharides, mucoidy, and/or buoyancy. However, BTLCPs are predicted to have signal sequences for export into the periplasm and are often encoded in the vicinity of trans-envelope transport systems or enzymes predicted to act in the periplasm, suggesting that they act on periplasmic or extra-cytoplasmic targets. Consistent with an enzymatic activity, mutations in predicted catalytic residues of HvyA abrogate function and we were able to demonstrate limited promiscuity among HvyA, NGR_c12490, and SMc00998 *in vivo*, at least under conditions of over-expression. Such protein crosstalk is not unexpected and has also been documented between non-cognate histidine kinases and response regulators for example. The crosstalk among the three transglutaminase/protease-like enzymes HvyA, NGR_c12490, and SMc00998 indicates similar targets in *Caulobacter* and *Sinorhizobia*. Unlike HvyA, NGR_c12490, and SMc00998, coding sequences are not embedded within genes encoding known surface polysaccharides export proteins. However, over-expression of HvyA did not alter the abundance or migration of the CPS export proteins CCNA_00162, CCNA_00163, CCNA_00164, CCNA_00167, and CCNA_00168 by SDS-PAGE immunoblotting, or the amount of assembled RsaA S-layer subunit (*Figure 2—figure supplement 3B*, *Figure 6—figure supplement 1*), indicating that HvyA is not required to maintain the steady state levels of these proteins. The distant HvyA relative LapG from *Pseudomonas fluorescens* cleaves the cell surface adhesin LapA that is encoded nearby in *P. fluorescens*, but not found in *Pseudomonas aeruginosa*, *Legionella pneumophila*, or *C. crescentus* genomes (*Newell et al., 2009*; *Navarro et al., 2011*; *Newell et al., 2011*; *Chatterjee et al., 2012*). While the target of the LapG ortholog from *L. pneumophila* is unknown, this enzyme can cleave *P. fluorescens* LapA *in vitro* (*Chatterjee et al., 2012*). LapG is regulated at the level of activity by sequestration to the cytoplasmic membrane by the cyclic-di-GMP responsive trans-membrane receptor LapD (*Navarro et al., 2011*). Moreover, the $Ca^{2+}$ ions not only promote outer membrane integrity, but also stimulate the cleavage of LapA by LapG in *P. fluorescens* (*Boyd et al., 2012*). Residues predicted to bind $Ca^{2+}$ ions are conserved in HvyA, and our unbiased structure–function analyses indeed unearthed mutations in conserved residues that are important for HvyA function. Thus, while control of HvyA activity by $Ca^{2+}$ ions could contribute to control of HvyA action on its proper targets within the envelope, the tight regulation of HvyA (and its orthologs) abundance during the alpha-proteobacterial cell cycle represents the major regulatory mechanism to constrain BTLCP activity temporally.

## Materials and methods

### Strains and growth conditions

*Caulobacter crescentus* NA1000 (*Evinger and Agabian, 1977*) and derivatives were grown at 30°C in PYE (peptone-yeast extract) or M2G (minimal glucose). *Sinorhizobium fredii* NGR234 (*Stanley et al., 1988*) was grown at 30°C in TY (tryptone-yeast extract). *Sinorhizobium meliloti* Rm2011 and derivatives

(*Casse, 1979*; *Becker et al., 1997*) were grown at 30°C in Luria broth (LB) supplemented with $CaCl_2$ 2.5 mM and $MgSO_4$ 2.5 mM. *Escherichia coli* S17-1 *λpir* (*Simon et al., 1983*), EC100D, EC100D *pir-116* (Epicentre Technologies, Madison, WI), and Rosetta(DE3)pLysS (Merck KGaA, Darmstadt, Germany) were grown at 37°C in LB. The *E. coli* mutator strain XL-1 Red (Agilent Technologies Inc., Cedar Creek, TX) was grown at 30°C in LB. Motility assays, swarmer cells isolation, electroporations, biparental matings, and bacteriophage φCr30-mediated generalized transductions were performed as described (*Ely, 1991*; *Viollier and Shapiro, 2003*; *Viollier et al., 2004*; *Chen et al., 2005*). Nalidixic acid, kanamycin, gentamicin, and tetracycline were used at 20 (8 for *S. meliloti*), 20, 1 (10 for *E. coli* and *S. meliloti*), and 1 (10 for *E. coli* and *S. meliloti*) µg/ml, respectively. Plasmids for β-galactosidase assays were introduced into *S. meliloti* by bi-parental mating and into *C. crescentus* by electroporation.

## Isolation of 'heavy' *himar1* (Tn) mutants

Transposon mutagenesis of Δ*pleC* and Δ*hvyA* strains was done by intergeneric conjugation from *E. coli* S17-1 *λpir* harbouring the *himar1*-derivative pHPV414 as previously described (*Viollier et al., 2004*). All the nalidixic acid and kanamycin resistant clones were pooled and grown in liquid medium before isolation of 'heavy' cells by centrifugation on density gradient. The isolated 'heavy' cells were subjected to three subsequent steps (of growth in liquid medium and centrifugation on density gradient) in order to obtain enrichment in 'heavy' mutant cells. After the third step of selection, 'heavy' cells were plated to obtain single colonies, and single Tn insertion sites were mapped by partial digestion of genomic DNA with *Hin*PI, religation, transformation of rescued plasmids into *E. coli* EC100D-*pir*116 and identification of insertion sites by sequencing as previously described (*Viollier et al., 2004*). The nucleotide positions of all mapped Tn insertions are reported in *Supplementary file 2*.

## Genome-wide transposon mutagenesis coupled to deep sequencing (Tn-Seq)

Transposon mutagenesis of *C. crescentus* NA1000 was done by intergeneric conjugation from *E. coli* S17-1 *λpir* harbouring the *himar1*-derivative pHPV414 (*Viollier et al., 2004*). Pools of Tn mutants of >100,000 kanamycin and nalidixic acid resistant clones were obtained for NA1000 grown either in the presence or absence of bacteriophage φCr30. To enrich the Tn pool for φCr30-resistant clones, after conjugation NA1000 cells were embedded into soft agar containing the appropriate antibiotics as well as φCr30 (at m.o.i. ≥2). After incubation of the plates at 30°C for 48 hr, all the clones were pooled for each Tn collection and chromosomal DNA was extracted. Sequencing (Illumina HiSeq 2000) and analysis were done as described previously (*Murray et al., 2013*).

## Production of antibodies and immunoblots

For the production of antibodies, His$_6$-SUMO-CCNA_00162$_{(51-422)}$, CCNA_00163$_{(101-300)}$-His$_6$, His$_6$-SUMO-CCNA_00164$_{(481-620)}$, His$_6$-HvyA$_{(26-272)}$, His$_6$-CCNA_00167$_{(1-108)}$, His$_6$-SUMO-CCNA_00168$_{(41-198)}$, and CCNA_02223$_{(22-289)}$-His$_6$ were expressed in *E. coli* Rosetta (DE3)pLysS cells and the recombinant proteins were purified using Ni-NTA agarose (Qiagen, Hilden, Germany). His$_6$-SUMO-CCNA_00168$_{(41-198)}$ and CCNA_02223$_{(22-289)}$-His$_6$ were purified in the soluble fraction and directly used to immunize rabbits (Josman LLC, Napa, CA). Purified His$_6$-SUMO-CCNA_00162$_{(51-422)}$, CCNA_00163$_{(101-300)}$-His$_6$, His$_6$-SUMO-CCNA_00164$_{(481-620)}$, His$_6$-HvyA$_{(26-272)}$, and His$_6$-CCNA_00167$_{(1-108)}$ were excised from 12.5% SDS polyacrylamide gels and used to immunize rabbits. For immunoblots, protein samples were separated on SDS polyacrylamide gel, transferred to polyvinylidene difluoride (PVDF) Immobilon-P membranes (Merck Millipore), and blocked in PBS (phosphate saline buffer), 0.1% Tween20, and 5% dry milk. The anti-sera were used at the following dilutions: anti-CtrA (1:10,000) (*Domian et al., 1997*), anti-PilA (1:10.000) (*Viollier et al., 2002*), anti-CcrM (1:10,000) (*Stephens et al., 1996*), anti-mCherry (1:10,000) (*Chen et al., 2005*), anti-CCNA_00162 (1:20,000), anti-CCNA_00163 (1:100,000), anti-CCNA_00164 (1:10,000), anti-HvyA (1:10,000), anti-CCNA_00167 (1:10,000), anti-CCNA_00168 (1:20,000), anti-β-lactamase (anti-Bla, 1:20,000). Protein-primary antibody complexes were visualized using horseradish peroxidase-labelled anti-rabbit antibodies and ECL detection reagents (Merck Millipore).

## Release of periplasmic proteins by EGTA

Cultures (8 ml) were grown to exponential phase (OD$_{600nm}$ ~ 0.6), centrifuged and washed twice with HEPES 10 mM pH 7.2. Cells were re-suspended in HEPES 10 mM pH 7.5 containing 10 mM EGTA (ethylene glycol tetraacetic acid) and incubated at room temperature for 10 min. Cells were then

pelleted by centrifugation and 20 µl of supernatant loaded on a 7.5% SDS polyacrylamide gel, followed by Coomassie Blue staining.

## HvyA stability analysis

*C. crescentus* cells expressing mCh-HvyA from the *hvyA* locus on the chromosome were grown in PYE to exponential phase ($OD_{600nm}$ ~ 0.6) before adding chloramphenicol (2 µg/ml). CtrA or mCh-HvyA levels were monitored by immunoblotting of samples taken at different time points after addition of chloramphenicol.

## Proteolysis of surface proteins

*C. crescentus* cells (*WT*, Δ*hvyA*, or Δ*CCNA_00163*) were grown in PYE to exponential phase ($OD_{600nm}$ ~ 0.6), pelleted by centrifugation, and re-suspended in 20 mM Tris, pH 7.5, 100 mM NaCl. The susceptibility of surface proteins to proteolysis was determined by treating whole cells with 0.5 mg/ml proteinase K; after incubation at 37°C (15, 30, 45, or 60 min), 1× protease inhibitors (Complete EDTA-free, Roche, Switzerland) were added. The cells were washed four times with 20 mM Tris (pH 7.5), 100 mM NaCl, 1× protease inhibitors, re-suspended in SDS-PAGE loading buffer and boiled. Protein samples were analysed by immunoblotting using antibodies to CCNA_00168.

## Site-directed mutagenesis of *hvyA*

In order to create *hvyA* alleles mutated in the predicted catalytic residues, the *hvyA* ORF was sub-cloned into pOK12 (*Vieira and Messing, 1991*) as *Nde*I/*Eco*RI fragment from pMT335-*hvyA*. pOK-*hvyA* was used as a template for oligonucleotide site-directed mutagenesis. Two complementary oligonucleotide primers containing the desired mutation were designed for each point mutation (*Supplementary file 5*). PCR reactions were composed of 30 cycles, carried out under the following conditions: denaturation, 94°C for 1 min; annealing, 60°C for 1 min; extension, 68°C for 8 min. The PCR products were treated with *Dpn*I to digest the template DNA and used to transform *E. coli* EC100D competent cells. The constructions obtained were verified by sequencing and sub-cloned as *Nde*I/*Eco*RI fragments into pMT335. In order to create TAP-tagged versions of the point mutants, the *hvyA* mutant alleles were amplified from pOK12 with hvyA_N and hvyA_CTIF primers and cloned into a pMT335 derivative harbouring the TAP epitope cloned as *Eco*RI/*Xba*I fragment (*Radhakrishnan et al., 2010*).

## Isolation of HvyA non-functional alleles

The HvyA loss-of-function alleles (R161P, W178S, D194G, H226Y, L240R, and P263R) were obtained following random mutagenesis of pUG52, which was passed through the *E. coli* mutator strain XL1-Red. The mutant pUG52 library was electroporated into *WT* NA1000 or the Δ*hvyA* strain. The transformants were pooled and subjected to centrifugation on density gradient, in order to isolate 'light' cells. These 'light' cells were grown in liquid medium and then subjected to three subsequent rounds of isolation by centrifugation on density gradient to obtain enrichment in 'light' cells. Cells isolated in the last density gradient were plated to obtain single colonies, from which we recovered the plasmid. Nine plasmids were sequenced and we identified six different mutations: clones L006 and L011 encoded the W178S variant, L009 encoded D194G, L101 encoded P263R, L102 and L110 encoded L240R, L104 and L111 encoded R161P, and L105 encoded H226Y.

## β-galactosidase assays

β-galactosidase assays were performed at 30°C. Cells (50–200 µl) at $OD_{660nm}$ = 0.1–0.5 were lysed with chloroform and mixed with Z buffer (60 mM $Na_2HPO_4$, 40 mM $NaH_2PO_4$, 10 mM KCl, and 1 mM $MgSO_4$, pH 7) to a final volume of 800 µl. 200 µl of ONPG (o-nitrophenyl-β-D-galactopyranoside, stock solution 4 mg/ml in 0.1 M potassium phosphate, pH 7) were added and the reaction timed. When a medium-yellow colour developed, the reaction was stopped by adding 400 µl of 1M $Na_2CO_3$. The $OD_{420nm}$ of the supernatant was determined and the Miller units (U) were calculated as follows: $U = (OD_{420nm} * 1000)/(OD_{660nm} * \text{time [in min]} * \text{volume of culture used [in ml]})$. Error was computed as standard deviation (SD).

## Capsule glycosyl composition analysis

*C. crescentus* cells grown in 2L PYE were pelleted by centrifugation, washed twice with phosphate saline buffer (PBS, pH 7.5), and lyophilized. Lyophilized cells were used for the purification of capsular

polysaccharides, which was performed using a modification of the method described by *Ravenscroft et al. (1991)* followed by glycosyl compositional analysis conducted by combined gas chromatography/mass spectrometry (GC/MS) of the per-*O*-trimethylsilyl (TMS) derivatives of the monosaccharide methyl glycosides produced by acidic methanolysis. Briefly, after treatment of cellular lysates with 95% ethanol for polysaccharide enrichment, contaminants such as DNA, RNA, and proteins were removed by successive digestion with DNase I, RNase A, and proteinase K. Every enzymatic digestion step was followed by dialysis against distilled deionized water. Samples were then subjected to ultracentrifugation (100,000×*g*, 18 hr, 4°C) to pellet lipopolysaccharide (LPS). The supernatant containing capsular polysaccharides was freeze-dried and used for glycosyl composition analysis. Inositol (20 µg) was added, as internal standard, to 500 µg of each sample. Polysaccharides were first hydrolysed with 2 M trifluoroacetic acid (TFA) at 120°C for 2 hr. Methyl glycosides were prepared from the dry samples by mild acid treatment (methanolysis in 1 M HCl in methanol at 80°C for 16 hr) followed by re-acetylation with pyridine and acetic anhydride in methanol (for detection of amino-sugars). The samples were then per-*O*-trimethylsilylated by treatment with Tri-Sil reagent (Thermo Scientific Pierce, Rockford, IL) at 80°C for 30 min (*York et al., 1985*; *Merkle and Poppe, 1994*). Gas chromatography/mass spectrometry (GC/MS) analysis of the TMS methyl glycosides was performed on an Agilent 7890A GC interfaced to a 5975C MSD, using an Agilent DB-1 fused silica capillary column (30 mm × 0.25 mm ID).

## Microscopy and fluorescence isothiocyanate (FITC)-dextran exclusion assay

Fluorescence and DIC imaging of *Caulobacter* cells were conducted as previously described (*Radhakrishnan et al., 2008*).

FITC-dextran with average mass of 2000 kDa (Sigma-Aldrich, St. Louis, MO) was used to assess capsule thickness as previously described (*Gates et al., 2004*). Briefly, cells were grown in PYE supplemented with 1% sucrose to a final $OD_{600nm}$ = 1.2. 500 µl of each culture were mixed, harvested by centrifugation at room temperature (3000×*g*, 5 min), washed once with PBS, and resuspended in 30 µl of PBS. 10 µl of bacterial suspension was mixed with 2 µl of FITC-dextran (10 mg/ml in water), applied onto a microscope slide, and firmly covered with a coverslip. Cells expressing SpmX-mCherry were grown in PYE supplemented with 1% sucrose to a final $OD_{600nm}$ = 0.6. SW and ST/PD cells were separated by centrifugation on density gradient, then washed with PBS, and incubated with FITC-dextran as described above. The samples were imaged as described for fluorescence and DIC images (*Radhakrishnan et al., 2008*). Images were analyzed with the MATLAB-based open-source software MicrobeTracker (*Sliusarenko et al., 2011*). Statistics were calculated using Graphpad Prism 4 and statistical significance was determined using a two-tailed Mann–Whitney test.

## AFM imaging of *Caulobacter* cells

*Caulobacter* cells grown overnight in liquid PYE were rinsed in PBS buffer and resuspended in 4% paraformaldehyde (Sigma-Aldrich) solution for 1 hr at room temperature for fixation. Cells were then rinsed in PBS buffer and filtered through polycarbonate porous membrane (Millipore, Billerica, MA, pore size: 3 µm). AFM imaging was performed using a Nanoscope VIII Multimode (Bruker Corporation, Santa Barbara, CA) and oxide-sharpened microfabricated $Si_3N_4$ cantilevers with a nominal spring constant of ~0.01 N/m (Microlevers, Veeco Metrology Group). After filtering the cell culture, the filter was gently rinsed with the buffer, carefully cut (1 cm × 1 cm), attached to a steel sample puck using a small piece of double face adhesive tape, and the mounted sample was transferred into the AFM liquid cell while avoiding dewetting. Images were taken in PBS buffer in contact mode under minimal applied force. Images were analysed using Nanoscope 8.10 software (Bruker, Santa Barbara, CA). Rms (root mean square) roughness values were calculated on 250 × 250 $nm^2$ areas of the high magnification height images subjected to second order filtering.

## Plasmid and strain construction

In-frame deletions and replacement of *hvyA* by a *mCherry-hvyA* N-terminal fusion (strain SA1737) were created using pNPTS138 derivatives constructed as follows:

pNPTS_*Δ00162*: PCR was used to amplify two DNA fragments flanking the *CCNA_00162* ORF, by using primers 162_ko1/162_ko2 and 162_ko3/162_ko4. The PCR fragments were digested with *Hind*III/*Bam*HI and *Bam*HI/*Eco*RI, respectively, then ligated into pNPTS138, restricted with *Hind*III and *Eco*RI.

pNPTS_Δ00163: PCR was used to amplify two DNA fragments flanking the *CCNA_00163* ORF, by using primers 163_ko1/163_ko2 and 163_ko3/163_ko4. The PCR fragments were digested with *Hind*III/*Bam*HI and *Bam*HI/*Eco*RI, respectively, then ligated into pNPTS138, restricted with *Hind*III and *Eco*RI.

pNPTS_Δ00164: PCR was used to amplify two DNA fragments flanking the *CCNA_00164* ORF, by using primers 164_ko1/164_ko2 and 164_ko3/164_ko4. The PCR fragments were digested with *Hind*III/*Bam*HI and *Bam*HI/*Eco*RI, respectively, then ligated into pNPTS138, restricted with *Hind*III and *Eco*RI.

pNPTS_Δ*hvyA*: PCR was used to amplify two DNA fragments flanking the *hvyA* ORF, by using primers hvyA_ko1/hvyA_ko2 and hvyA_ko3/hvyA_ko4. The PCR fragments were digested with *Eco*RI/*Bam*HI and *Bam*HI/*Hind*III, respectively, then ligated into pNPTS138, restricted with *Hind*III and *Eco*RI.

pNPTS_P$_{hvyA}$-*mCh::hvyA*: primers hvyA_up_H and hvyA_up_B were used to amplify a 921-bp fragment encompassing the region upstream of *hvyA* and the first 78 bp of the *hvyA* ORF (encoding the signal sequence). Primers mCh_B and mCh_X were used to amplify the mCherry coding sequence (without ATG and stop codon). Primers hvyA_down_X and hvyA_down_E were used to amplify a 719-bp fragment of *hvyA* ORF. The three PCR fragments were digested with *Hind*III/*Bam*HI, *Bam*HI/*Xba*I, and *Xba*I/*Eco*RI, respectively, and ligated into pNPTS138, restricted with *Hind*III and *Eco*RI.

pNPTS_Δ00167: PCR was used to amplify two DNA fragments flanking the *CCNA_00167* ORF, by using primers 167_ko1/167_ko2 and 167_ko3/167_ko4. The PCR fragments were digested with *Eco*RI/*Bam*HI and *Bam*HI/*Hind*III, respectively, then ligated into pNPTS138, restricted with *Hind*III and *Eco*RI.

pNPTS_Δ00167$_{(ΔhvyA)}$: in order to obtain an in-frame deletion of *CCNA_00167* in the Δ*hvyA* background, primers 167_ko5/167_ko6 were used to amplify by PCR a 1085-bp fragment upstream of *CCNA_00167* (using genomic DNA of the Δ*hvyA* strain as template). The PCR fragment was digested with *Eco*RI/*Bgl*II and ligated into pNPTS_Δ00167, restricted with *Bam*HI and *Eco*RI.

pNPTS_Δ03998: PCR was used to amplify two DNA fragments flanking the *CCNA_03998* ORF, by using primers 3998_ko1/3998_ko2 and 3998_ko3/3998_ko4. The PCR fragments were digested with *Eco*RI/*Bam*HI and *Bam*HI/*Hind*III, respectively, then ligated into pNPTS138, restricted with *Hind*III and *Eco*RI.

pNPTS_Δ00466: PCR was used to amplify two DNA fragments flanking the *CCNA_00466* ORF, by using primers 466_ko1/466_ko2 and 466_ko3/466_ko4. The PCR fragments were digested with *Mun*I/*Bam*HI and *Bam*HI/*Hind*III, respectively, then ligated into pNPTS138, restricted with *Hind*III and *Eco*RI.

pNPTS_Δ00467: PCR was used to amplify two DNA fragments flanking the *CCNA_00467* ORF, by using primers 467_ko1/467_ko2 and 467_ko3/467_ko4. The PCR fragments were digested with *Hind*III/*Bam*HI and *Bam*HI/*Eco*RI, respectively, then ligated into pNPTS138, restricted with *Hind*III and *Eco*RI.

pNPTS_Δ00470: PCR was used to amplify two DNA fragments flanking the *CCNA_00470* ORF, by using primers 470_ko1/470_ko2 and 470_ko3/470_ko4. The PCR fragments were digested with *Eco*RI/*Bam*HI and *Bam*HI/*Hind*III, respectively, then ligated into pNPTS138, restricted with *Hind*III and *Eco*RI.

Bi-parental matings were used to transfer the resulting pNPTS138 derivatives into *C. crescentus* strains. Double recombination was selected by plating bacteria onto PYE plates containing 3% sucrose. Putative mutants were confirmed by PCR using primers external to the DNA fragments used for the pNPTS138 constructs.

To inactivate the *rsaA* gene in the Δ*hvyA*, Δ*CCNA_00163*, or Δ*hvyA* Δ*CCNA_00163* mutant strains, plasmid pNPTS138_Δ*rsaA* was introduced into the strains by bi-parental mating. Clones that had undergone a single recombination event were selected on PYE plates containing kanamycin and verified by PCR.

To created strain SA1984 (Δ*mucR1*Δ*mucR2* with *hvyA* replaced by N-terminal mCherry-tagged *hvyA*), a 719-bp fragment was amplified by PCR using primers hvyA_in_B and hvyA_E. The PCR fragment was digested with *Bam*HI/*Eco*RI and ligated into pGS18T, restricted with the same enzymes. The resulting plasmid (pSA480) was integrated into the *hvyA* locus in SA1737 (strain SA1951), and the *mCh-hvyA* fusion was transduced into the Δ*mucR1*Δ*mucR2* strain by φCr30-mediated transduction and selection on PYE kanamycin plates.

To create the P$_{hvyA}$-hvyA::lacZ translational fusion (pSA184), a 531-bp DNA fragment, encompassing 513-bp upstream of hvyA and the first six codon of hvyA ORF, was amplified by PCR with primers PhvyA_B/PhvyA_P, digested with BglII and PstI, and ligated into pJC327 (**Chen et al., 2006**), restricted with the same enzymes.

To create the P$_{hvyA}$-lacZ transcriptional fusion (pSA205), the fragment corresponding to the hvyA promoter region was excised from pSA184 with BglII and PstI, and ligated into pRKlac290 (**Gober and Shapiro, 1992**), cut with BamHI and PstI.

To create the P$_{SMc00998}$-lacZ transcriptional fusion (pSA146), a 542-bp DNA fragment was amplified by PCR with primers Sm998_B/Sm998_P from S. meliloti genomic DNA, digested with BamHI and PstI, and ligated into pRKlac290, cut with the same enzymes.

Plasmids for β-galactosidase assays were introduced into S. meliloti Rm2011 and Rm101 by bi-parental mating.

To complement the ΔhvyA mutation, the hvyA ORF was amplified by PCR with primers hvyA_N and hvyA_E. The resulting PCR product was digested with NdeI and EcoRI and ligated into pMT335 (P$_{van}$, medium copy plasmid; pMT335-hvyA) or pMT375 (**Thanbichler et al., 2007**) (P$_{xyl}$, low copy plasmid; pMT375-hvyA), restricted with the same enzymes.

To create the P$_{van}$-hvyA-TAP fusion (plasmid pUG52), the hvyA ORF was amplified by PCR with primers hvyA_N and hvyA_CTIF (without stop codon). The PCR fragment was digested with NdeI and EcoRI and cloned into a pMT335 derivative harbouring the TAP epitope cloned as EcoRI/XbaI fragment (**Radhakrishnan et al., 2010**).

Plasmids to complement the in-frame deletion mutants and for over-expression (from P$_{van}$) were constructed as follows:

pSA362: CCNA_00162 ORF was amplified by PCR with primers 162_N (with NdeI site overlapping the start codon) and 162_M (with MunI site flanking the stop codon) and cloned into pMT335, restricted with NdeI and EcoRI.

pSA361: CCNA_00163 ORF was amplified by PCR with primers 163_N (with NdeI site overlapping the start codon) and 163_E (with EcoRI site flanking the stop codon) and cloned into pMT335, restricted with NdeI and EcoRI.

pSA401: CCNA_00164 ORF was amplified by PCR with primers 164_N (with NdeI site overlapping the start codon) and 164_E (with EcoRI site flanking the stop codon) and cloned into pMT335, restricted with NdeI and EcoRI.

pSA62: CCNA_00167 ORF was amplified by PCR with primers 167_N (with NdeI site overlapping the start codon) and 167_E (with EcoRI site flanking the stop codon) and cloned into pMT335, restricted with NdeI and EcoRI.

pSA324: CCNA_00168 ORF was amplified by PCR with primers 168_N (with NdeI site overlapping the start codon) and 168_E (with EcoRI site flanking the stop codon) and cloned into pMT335, restricted with NdeI and EcoRI.

pUG35: CCNA_03998 ORF was amplified by PCR with primers 3998_N (with NdeI site overlapping the start codon) and 3998_E (with EcoRI site flanking the stop codon) and cloned into pMT335, restricted with NdeI and EcoRI.

pUG28: CCNA_00466 ORF was amplified by PCR with primers 466_N (with NdeI site overlapping the start codon) and 466_M (with MunI site flanking the stop codon) and cloned into pMT335, restricted with NdeI and EcoRI.

pSA102: CCNA_00470 ORF was amplified by PCR with primers 470_N (with NdeI site overlapping the start codon) and 470_E (with EcoRI site flanking the stop codon) and cloned into pMT335, restricted with NdeI and EcoRI.

pSA264: SMc00998 ORF was amplified by PCR from S. meliloti genomic DNA with primers Sm998_N (with NdeI site overlapping the start codon) and Sm998_E (with EcoRI site flanking the stop codon) and cloned into pMT335, restricted with NdeI and EcoRI.

pSA142: NGR_c12490 ORF was amplified by PCR from S. fredii NGR234 genomic DNA with primers Sf12490_N (with NdeI site overlapping the start codon) and Sf12490_M (with MunI site flanking the stop codon) and cloned into pMT335, restricted with NdeI and EcoRI.

pSA141: NGR_c19800 ORF was amplified by PCR from S. fredii NGR234 genomic DNA with primers Sf19800_N (with NdeI site overlapping the start codon) and Sf19800_E (with EcoRI site flanking the stop codon) and cloned into pMT335, restricted with NdeI and EcoRI.

pSA147: *NGR_c36180* ORF was amplified by PCR from *S. fredii* NGR234 genomic DNA with primers Sf36180_N (with *Nde*I site overlapping the start codon) and Sf36180_E (with *Eco*RI site flanking the stop codon) and cloned into pMT335, restricted with *Nde*I and *Eco*RI.

pSA309: *Atu0252* ORF was amplified by PCR from *A. tumefaciens* genomic DNA with primers At252_N (with *Nde*I site overlapping the start codon) and At252_E (with *Eco*RI site flanking the stop codon) and cloned into pMT335, restricted with *Nde*I and *Eco*RI.

pMT335-*Bh_MucR*: a synthetic fragment encoding the *mucR* homolog from *Bartonella henselae* (*PRJBM_00467*) was ligated into pMT335 (using *Nde*I/*Eco*RI).

Synthetic DNA fragment (Integrated DNA Technologies) encoding *PRJBM_00467* (codon optimized for *C. crescentus*) (5'–3'):

*CATATG*GAGCACCGACCGGTGCTGGAAACCGAGTCGAATCTGGTCATCACCCTCGTCGCCGAC
ATCGTCGCCGCGTATGTGTCGAACAACTCCATCCGTCCCACCGAGGTCCCCAGCCTCATCGCGGA
CGTCCATGCGGCTTTCCGCAAGGCCGGCAACGCCGACTTGACGGAAGTTGAGGTGGAGAAGCA
GCGCCCTGCGGTCAACCCGAAGCGCAGCATCTTCCCGGACTACCTTATCTGCCTGGAAG
ATGGCAAGAAGTTCAAGAGCCTGAAGCGCCACCTGATGACGCACTATGGCATGCTGCCGG
AAGAGTATCGCGAGAAGTGGCAGCTGGACTCTTCGTACCCCATGGTGGCCCCGAACTA
CGCGAAGGCCCGGTCGGCCCTGGCCAAAGAGATGGGCCTGGGGCGGAAGTCCAAGC
GGAAAAAGACCAAGT*GAATTC*

Plasmid pSA354 is a derivative of pCWR547 (*Radhakrishnan et al., 2010*) expressing His$_6$-SUMO-CCNA_00162$_{(51-422)}$ under the control of the T7 promoter. To construct pSA354, a fragment encoding residues 51-422 of CCNA_00162 was amplified by PCR with primers 162_in_N and 162_in_S, digested with *Nde*I and *Sac*I, and cloned into pCWR547, restricted with the same enzymes.

Plasmid pCWR508 is a derivative of pET-47b (Novagen) expressing CCNA_00163$_{(101-300)}$-His$_6$ under the control of the T7 promoter. To construct pCWR508, a fragment encoding residues 101-300 of CCNA_00163 was amplified by PCR with primers 163_in_N and 163_His_E (that also encodes six His residues followed by a stop codon and Eco*RI* site), digested with *Nde*I and *Eco*RI, and cloned into pET47b, restricted with the same enzymes.

Plasmid pSA352 is a derivative of pCWR547 expressing His$_6$-SUMO-CCNA_00164$_{(481-620)}$ under control of the T7 promoter. To construct pSA352, a fragment encoding residues 481-620 of CCNA_00164 was amplified by PCR with primers 164_in_N and 164_in_S, digested with *Nde*I and *Sac*I, and cloned into pCWR547, restricted with the same enzymes.

Plasmid pET-*hvyA* is a derivative of pET-28a (Novagen) expressing His$_6$-HvyA$_{(26-272)}$ under the control of the T7 promoter. To construct pET-hvyA, a fragment encoding residues 26-272 of HvyA was amplified by PCR with primers hvyA_short and hvyA_E, digested with *Nde*I and *Eco*RI and cloned into pET-28, restricted with the same enzymes.

Plasmid pET-*00167* is a derivative of pET-28a expressing His$_6$-CCNA_00167$_{(1-108)}$ under the control of the T7 promoter. To construct pET-00167, a fragment encoding residues 1-208 of CCNA_00167 was amplified by PCR with primers 167_N and 167_in_E, digested with *Nde*I and *Eco*RI, and cloned into pET-28, restricted with the same enzymes.

Plasmid pSA342 is a derivative of pCWR547 expressing His$_6$-SUMO-CCNA_00168$_{(41-198)}$ under the control of the T7 promoter. To construct pSA342, a fragment encoding residues 41-198 of CCNA_00168 was amplified by PCR with primers 168_short and 168_E, digested with *Nde*I and *Eco*RI, and cloned into pET-28. The CCNA_00168 fragment was then sub-cloned into pCWR547 using *Nde*I/*Sac*I.

Plasmid pCWR496 is a derivative of pET-47b expressing CCNA_02223$_{(22-289)}$-His$_6$ under control of the T7 promoter. To construct pCWR496, a fragment encoding residues 22-289 of CCNA_02223 (β-lactamase) was amplified by PCR with primers bla_N and bla_His_E (that also encodes six His residues followed by a stop codon and *Eco*RI site), digested with *Nde*I and *Eco*RI, and cloned into pET47b, restricted with the same enzymes.

## Acknowledgements

Support is from SNF grant #31003A_143660 and grant A/C ME9175 from the Synapsis Foundation and the Stammbach-Stiftung to PV. Work at the Université catholique de Louvain was supported by

the National Fund for Scientific Research (FNRS), the Federal Office for Scientific, Technical and Cultural Affairs (Interuniversity Poles of Attraction Programme), and the Research Department of the Communauté française de Belgique (Concerted Research Action), Y.F.D. is a Research Director of the FNRS. We thank Melissa Marks and Urs Jenal for communicating results prior to publication and Yves Brun for sending the pNPTS_ΔpssY plasmid. We thank Antonio Frandi, Haleh Yasrebi, and Gaël Panis for their help with the Tn-seq analysis. This research was supported in part by the Department of Energy grant Plant and Microbial Complex Carbohydrates (DE-FG02-93ER20097) to Parastoo Azadi at the Complex Carbohydrate Research Center.

## Additional information

### Funding

| Funder | Grant reference number | Author |
| --- | --- | --- |
| Swiss National Science Foundation | 31003A_143660 | Patrick H Viollier |
| Synapsis Foundation | Stammbach Stiftung A/C ME9175 | Patrick H Viollier |

The funders had no role in study design, data collection and interpretation, or the decision to submit the work for publication.

### Author contributions

SA, PHV, Conception and design, Acquisition of data, Analysis and interpretation of data, Drafting or revising the article; CF, MB, SKR, Conception and design, Acquisition of data, Analysis and interpretation of data; AB, Acquisition of data, Analysis and interpretation of data; LT, Contributed to all molecular biology steps (e.g. cloning), construction of mutant strains and beta-galactosidase assays, Acquisition of data; YFD, Conception and design, Analysis and interpretation of data

## Additional files

### Supplementary files

• Supplementary file 1. Phenotypes of *WT Caulobacter* and mutant strains.

• Supplementary file 2. Nucleotide position of *himar1* insertions.

• Supplementary file 3. Strains used in this study.

• Supplementary file 4. Plasmids used in this study.

• Supplementary file 5. Oligonucleotides used in this study.

• Supplementary file 6. Global Tn-insertion values for the Tn-Seq experiment.

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
