## [Decision Letter]

Thank you for sending your work entitled “Cell cycle constraints on capsulation and genetic exchange” for consideration at *eLife.* Your article has been favorably evaluated by Vivek Malhotra (Senior editor), a Reviewing editor, and 3 reviewers, one of whom, Ry Young, has agreed to reveal his identity (the other 2 remain anonymous).

The Reviewing editor and the other reviewers had a vigorous discussion of the implications of your findings before reaching the decision to consider your manuscript for publication. The Reviewing editor has assembled the following comments to help you prepare a revised submission.

In this paper, Ardissone et al. investigate the molecular basis of the cell cycle switch in cellular buoyancy in Caulobacter crescentus and doing so, demonstrate that it is linked to cell capsulation and its control during the cell cycle. Specifically, the authors show that the expression of HvyA, a newly identified negative regulator of capsulation is both under tight transcriptional and translational cell cycle control to become cleared when the cells enter the S-phase, allowing the capsulation of a particular cell type. This ensures a cell-specific protection against phage infections and potentially other environmental insults. The regulatory mechanism has been genetically dissected and shown to involve known cell cycle regulators MucR1/2, SciP, CtrA and a still elusive factor X. This regulation or at least the MucR part may be conserved in other alphas. HvyA inhibits capsule formation, but its mode of action, which appears to be conserved in other alpha proteobacteria, remains unknown. Overall, the experiments are high quality and the conclusions are justified.

However, all reviewers agree the following modifications are critical:

1) The text needs to be condensed and revised to make the paper easier to read. In addition, the discussion that capsule was selected for phage depredation should be eliminated. Instead, the authors should discuss other possible benefits of the capsule and the observed regulation.

2) Since “seeing is believing”, the authors should image the capsule at the surface of the stalk (possibly by EM) in both wild type and mutant strains.

---

## [Author Response]

The revised manuscript features the following editorial and experimental amendments.

1) We streamlined the text, with a shortened Introduction and Discussion as recommended.

2) We included cytological experiments providing compelling evidence that Caulobacter S-phase cells are encased in a polysaccharidic capsule.

2a) First, negative staining fluorescence microscopy revealed a thick layer that excludes FITC-labeled dextran from the delta-*hvyA* (capsulated) mutant versus the delta-*CCNA00163* (non-capsulated) mutant in which a component of the export machinery has been inactivated. These fluorescence images and quantification are show in the new Figure 4.

2b) Second, atomic force microscopy of the same strains revealed a smoother surface on delta-*CCNA00163* cells compared to delta-*hvyA* cells (new Figure 5). The difference in surface roughness is consistent with the biophysical properties expected for an amorphous layer of polysaccharides.

2c) Third, we used live cell fluorescence microscopy of cells expressing a different component of the capsule export machinery (CCNA00162, CCNA00163 and CCNA00168) as full-length translational fusion to mCherry to find that the mCherry-derived fluorescence emanates through the cell, suggesting that the export machinery (and thus the exported capsule) is not localized to specific subcellular site. These images are shown in Figure 4–figure supplement 1).

2d) Lastly, negative stain fluorescence microscopy of Caulobacter (NA1000) WT cells using FITC-dextran revealed that S-phase (stalked and/or constricted) cells exclude the dye. Cells that do not exclude the dye are stalkless and non-constricted (and thus in G1-phase). This important result that reinforces our conclusion that G1-cells are non-capsulated is also shown in the new Figure 4.